# The complex genetic landscape of familial MDS and AML reveals pathogenic germline variants

Ana Rio-Machin ⓘ et al.#

The inclusion of familial myeloid malignancies as a separate disease entity in the revised WHO classification has renewed efforts to improve the recognition and management of this group of at risk individuals. Here we report a cohort of 86 acute myeloid leukemia (AML) and myelodysplastic syndrome (MDS) families with 49 harboring germline variants in 16 previously defined loci (57%). Whole exome sequencing in a further 37 uncharacterized families (43%) allowed us to rationalize 65 new candidate loci, including genes mutated in rare hematological syndromes (*ADA, GP6, IL17RA, PRF1* and *SEC23B*), reported in prior MDS/AML or inherited bone marrow failure series (*DNAH9, NAPRT1* and *SH2B3*) or variants at novel loci (*DHX34*) that appear specific to inherited forms of myeloid malignancies. Altogether, our series of MDS/AML families offer novel insights into the etiology of myeloid malignancies and provide a framework to prioritize variants for inclusion into routine diagnostics and patient management.

#A full list of authors and their affiliations appears at the end of the paper.

nherited forms of myeloid malignancies are thought to be rare, although the precise incidence and prevalence are not known. The inclusion of familial myeloid malignancies as a separate disease entity in the World Health Organization (WHO) classification of hematological cancers in 2016[1] and their recognition by the European Leukaemia Net (ELN)[2] has led to greater awareness of the existence of these heritable forms of disease. It also highlights the importance of early diagnosis and tailored management of this group of patients and families[3]. Not only can families benefit from appropriate genetic counseling and long-term surveillance, there are several examples reported in the literature of donor-derived malignancy following transplant[4–6] from an asymptomatic family member and the benefit of modifying conditioning regimen in patients with an underlying telomerase mutation prior to bone marrow transplant[7].

The clinical recognition of familial transmission is challenging as family sizes tend to be small with incidence frequently masked by marked differences in disease latency and phenotype, exacerbated by the absence of routine customized familial diagnostic testing. The problem is more acute in the adult population compared with pediatric onset of disease, where there can be a heightened suspicion on behalf of the treating physician and where a malignancy can arise as part of a wider syndrome[3]. Many patients/families initially exhibiting bone marrow failure syndromes such as Fanconi anemia, dyskeratosis congenita and Shwachman–Diamond syndrome are prone to the development of myelodysplastic syndrome (MDS) or acute myeloid leukemia (AML)[8].

It is 20 years since inherited variants in the transcription factor RUNX1 were identified as being responsible for familial platelet disorder with a propensity to develop AML (FPD-AML)[9]. Reflecting the uptake of next-generation sequencing approaches, we are now aware of more than 20 genes linked with heritable myeloid malignancies[8] (Table 1), with a significant number of newly described loci appearing restricted to a few core families (ATG2B/GSK1P[10], ETV6[11,12], SRP72[13]. Several of these familial loci are also mutated in sporadic AML/MDS (RUNX1[9], CEBPA[14], GATA2[15]) while others appear enriched or exclusive to inherited forms of myeloid malignancies (DDX41 [16]). These examples offer a unique insight into the underlying etiology of myeloid malignancies and present opportunities to understand disease latency, penetrance and the reasons behind the striking intra- and inter-clinical heterogeneity that is a feature of many of these families.

To address this issue, we have analyzed the largest series of AML/MDS families assembled to date. It exemplifies the molecular heterogeneity of inherited AML and MDS[17], with likely pathogenic variants in 16 previously defined loci, detected in 49 (57%) families. Whole exome sequencing (WES) in a further 37 "uncharacterized" families (43%) focus attention on 144 variants in 65 candidate genes, 26 (40%) of which are mutated in cases of sporadic AML. This includes genes mutated in rare hematological syndromes (ADA, GP6, IL17RA, PRF1, and SEC23B), reported in prior MDS/AML or inherited bone marrow failure (IBMF) series (DNAH9, SH2B3, and NAPRT1), or novel loci where loss-of-function of the identified germline variants was demonstrated (DHX34).

## Results

**Cohort of familial AML/MDS cases.** In this study, "familial AML/MDS" references families where two or more family members (usually first-degree relatives) were diagnosed with a hematological disorder (predominantly AML, MDS, aplastic anemia, or thrombocytopenia) with at least one of the affected cases being categorized as MDS or AML. Bone marrow, peripheral blood or other non-hematopoietic samples were collected from at least one affected individual from 86 families (FML001-086) and the corresponding laboratory and clinical information was collated to capture relevant information relating to patients and their relatives (Table 2; Supplementary Figs. 1 and 2). Our cohort of cases has not been accrued as part of any systematic initiative, but instead, has been based on ad hoc referral of cases, which has spanned several years, from both UK and international institutions. Across our series of 86 families, we have collated 297 samples corresponding to 221 individuals, of which 168 were affected and 53 were unaffected (Supplementary Data 1). In 33 families, material was available on the index case only. In five additional cases, we have samples on the index case and at least one additional unaffected individual. In the remaining 48 families, our collection includes samples from multiple affected individuals allowing us to perform segregation analysis on identified variants. For simplicity, we have assigned our families into two discrete groups (Table 1) based on the panel used for clinical indication "R347 Inherited predisposition to acute myeloid leukemia (AML)" in the NHS England Genomic Medicine Service [https://www.england.nhs.uk/publication/national-genomic-test-directories/]. Group 1, where a variant has been detected either at a locus with high/moderate level of evidence for gene–disease association, is emerging from basic research or is mutated in other inherited hematological syndromes (FML001-FML049; Fig. 1; Supplementary Fig. 1; Supplementary Data 2); Group 2, where the molecular lesion has not been defined (FML050–FML086; Supplementary Fig. 2; Supplementary Data 3).

**Group 1 families with variants in established genes.** Of the 49 Group 1 families (Fig. 1; Supplementary Fig. 1; Supplementary Data 2), 32 have been reported as part of previously published studies[9,13,14,18–32]. The additional 17 families (Fig. 2; Table 3; Supplementary Data 4) harbor variants in 13 discrete loci and together highlight the high index of suspicion needed on behalf of the treating physician to infer an inherited malignancy which warrant further investigation. The clinical heterogeneity is reflected in variations in age of onset from 1 to 68 years, MDS/AML co-occurring with other non-hematological features in six families (FML019, FML037, FML041, FML044, FML045, and FML047) and variable penetrance observed in four families (FML007, FML014, FML018, and FML041) (Fig. 2; Table 3; Supplementary Data 2).

Family FML007 represents the second example of a C-terminal CEBPA germline variant, p.Gln330Argfs*74, that is associated with reduced penetrant (over 50% of carriers are asymptomatic) AML (Fig. 2a). This is in contrast to the more conventional near

**Table 1 Genetic classification of MDS/AML families.**

| Group | Description | Genes | | Number of families |
|---|---|---|---|---|
| Group 1 | Families with variants in known genes | High level of evidence for gene–disease association | ANKRD26, CEBPA, DDX41, ETV6, GATA2, RUNX1, TERC, TERT, TP53 | 39 (45%) |
| | | Moderate level of evidence for gene–disease association | ACD, CHEK2, RTEL1, SAMD9, SAMD9L, SRP72 | 4 (5%) |
| | | Genes emerging from basic research or mutated in other inherited hematological syndromes with high risk of MDS/AML | ATG2B/GSKIP, ERCC6L2, FANC genes, MBD4, MECOM, SBDS, WAS | 6 (7%) |
| Group 2 | Families where the specific gene variants have not been established | Unknown | | 37 (43%) |

**Table 2 Summary of the cohort of MDS/AML families.**

| Characteristics | Number of families (%) |
|---|---|
| Clinical phenotype | |
| MDS | 12 (14) |
| AML | 17 (19.8) |
| MDS/AML | 18 (20.9) |
| MDS/AML/TCP | 15 (17.4) |
| MDS/AML/BMF | 24 (27.9) |
| Family history of hematological disease | 86 (100) |
| Consanguinity | 3 (3.5) |
| Other malignancies/disorders | |
| Non-hematological cancers | 12 (14) |
| Neurological disorders | 7 (8.1) |
| Other hematological malignancies (CML, ALL, non-Hodgkin's lymphoma) | 3 (3.5) |
| Pulmonary fibrosis | 2 (2.3) |
| Liver disease | 2 (2.3) |

fully penetrant N-terminal variants[33], seen in most examples of pure familial *CEBPA* AML. *DDX41* variants were detected in probands from FML012 (p.Met1?) and FML013 (p.Arg124*) (Fig. 2b, c), with both affected patients diagnosed with MDS at 68 years. The heritable presentation in this family could be easily overlooked in favor of sporadic disease due to its unusual late onset[17]. The heterozygous germline *GATA2* variant p.Thr354Met, present in both the index case (III.1) and his asymptomatic father (II.2) of family FML018 (Fig. 2e), represents a further example of reduced penetrance linked to this particular missense *GATA2* variant, where variable penetrance has been associated with fluctuations in the expression of the mutated allele[34]. The occurrence of MDS/AML in certain families may also coincide with other non-hematological features: the index case of family FML019 presented with MDS and congenital deafness, analogous to a case reported by Saida et al.[35] in a family with identical germline *GATA2* variant (p.Arg362*) (Fig. 2f). In family FML037, harboring a novel *TERC* variant n.179_180delinsGG (absent in ExAC database) (Fig. 2l), both affected individuals (II.2 and II.1) were prematurely gray in their 20s, a feature frequently associated with *TERC* and *TERT* variants. Our series also includes three examples of FPD-AML (FML029, FML030, and FML031) where sequencing analysis failed to detect any *RUNX1* variants (Fig. 2g–i; Supplementary Data 2). Familial platelet disorder with predisposition to myeloid malignancy (FPDMM) is normally associated with germline *RUNX1* variants and characterized by life-long moderate thrombocytopenia and platelet function defects with heterogeneity both in age and presentation and significant risk of MDS/AML transformation, estimated at 20–60%[8]. Although most of the FPDMM families harbor germline *RUNX1* variants, inherited large intragenic duplications or deletions have been also described in these families[36–39]. Therefore, we performed a subsequent profiling to detect large copy number variants and we identified novel germline *RUNX1* deletions in the three families with FPD-AML. A *RUNX1* deletion (chr21:36349450–36572837) was detected in patient II.5 of family FML029 (Fig. 2h; Supplementary Fig. 3a) and confirmed in other affected family members (III.2, III.3, and III.4) (Supplementary Fig. 3b). In family FML030, a *RUNX1* deletion was observed in II.2 that was absent in his healthy sister (II.1) (Fig. 2g; Supplementary Fig. 3). Peripheral blood and salivary DNA from II.2 of family FML031 revealed a *RUNX1* 666 kb deletion (chr21: 36389457–37055677) (Fig. 2i; Supplementary Fig. 3). Altogether, a comparison of breakpoints in the three families revealed a "hotspot" for constitutional deletion, with a common deleted

segment spanning ~100 kb (chr21: 36400658–36572837) that encompasses the distal *RUNX1* promoter and first two exons. These results highlight the importance of performing copy number assessment in families or individuals who present with an appropriate clinical phenotype.

**Identifying new candidate genes in Group 2 families.** We performed WES analysis in the remaining 37 families (FML050–FML086; EGAD00001004539; Supplementary Fig. 2; Supplementary Data 3) where we had failed to detect a mutation in any of the Group 1 genes. It is worth noting that we cannot rule out the possibility that our discovery series includes cases of known loci where CNVs or non-coding regulatory mutations represent novel molecular mechanisms which would go undetected using a coding-only screening strategy, or indeed the co-occurrence of independent sporadic cases of disease in the same family. In Group 2, we did not include families where we had assigned a variant to a category of "variant of unknown significance" (VUS). In FML014, for example (Fig. 2d), the index case (IV.2) retained a novel heterozygous germline *ETV6* variant c.349C>T (p. Leu117Phe) with low ExAC frequency (8.237E-06) but samples from other affected family members (maternal grandfather (II.2), maternal great aunt (II.3), and maternal great-grandmother (I.2)) who had all died of leukemia were unavailable. This exemplifies the inherent problems in classifying variants as well as searching for new loci in families where the molecular lesion has not been described or when material is limited and there is too great a reliance on the index case.

Our WES data suggest that there is no single gene that can account for the large number of unattributed heritable cases in our series. We observed on average 311 rare single-nucleotide variants (minor allele frequency <0.005, range 161–781) per patient. In order to identify variants with a causative role in familial MDS/AML, we applied the following criteria (Supplementary Fig. 4): (i) demonstrated variant allele frequencies >30%, (ii) indels (6.2%), nonsense (2.4%), or splicing (±1, 2, or 3) (5.1%) variants affecting the canonical transcript, (iii) segregated with the disease where testing was possible, (iv) a frequency of <0.0001 in population controls (ExAC database), (v) predicted to be pathogenic with two out of four tools used for functional annotation (PolyPhen2, MutationTaster, SIFT and Provean), and (vi) detected in the same gene in at least two families. In order to mitigate the large number of missense variants (86.3%), genes were only included if missense variants were detected in three or more families. These criteria defined 130 variants in 52 candidate genes (Supplementary Data 5). We also took advantage of other published series, cross-referencing our WES data with the familial MDS/AML candidate loci selected by Churpek et al.[40], the potential candidate gene in the MDS/AML family described by Patnaik et al.[41] and the causal, likely causal or possibly contributing genes identified in a cohort of inherited BMF patients by Bluteau et al.[42] (Supplementary Data 6). This has allowed us to include an additional 13 genes, overlapping with these previous studies (Supplementary Data 7).

The combination of both criteria resulted in a final list of 144 variants in 65 genes across 34 of the 37 families (Supplementary Fig. 5; Supplementary Data 8). Multiple variants were detected in the majority of families (30/37) with no variant observed in FML052, 060, and 066. All variants selected for Sanger sequencing were validated ($n = 48$) and segregation analysis confirmed the germline transmission in all cases tested where material was available from more than one affected individual (Supplementary Fig. 6). Consistent with the germline origin, the median VAF of all 144 variants was 0.48. Twenty-six (40%) of the 65 genes were previously reported to be mutated in at least one example of sporadic AML from

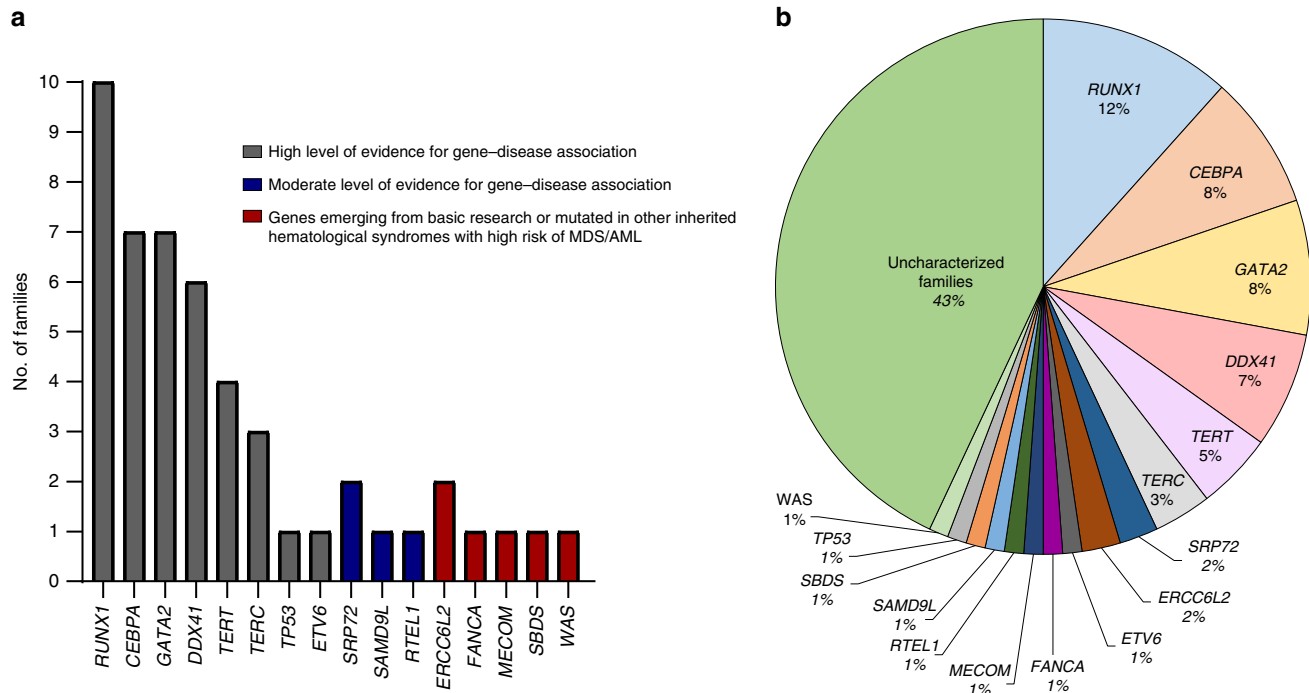

**Fig. 1 Group 1 families: Families with variants in established genes. a** Number of families with variants in known disease-causing loci. In gray, genes where the level of evidence for gene–disease association is high; moderate level of evidence in blue; and genes emerging from basic research or mutated in other inherited hematological syndromes with high risk of MDS/AML in red. Variants in these genes are heterozygous, apart from *ERCC6L2*, *FANCA*, and *SBDS* where they are biallelic, and *WAS* where it is hemizygous. **b** Percentages of our family cohort with causal variants in known disease-causing genes.

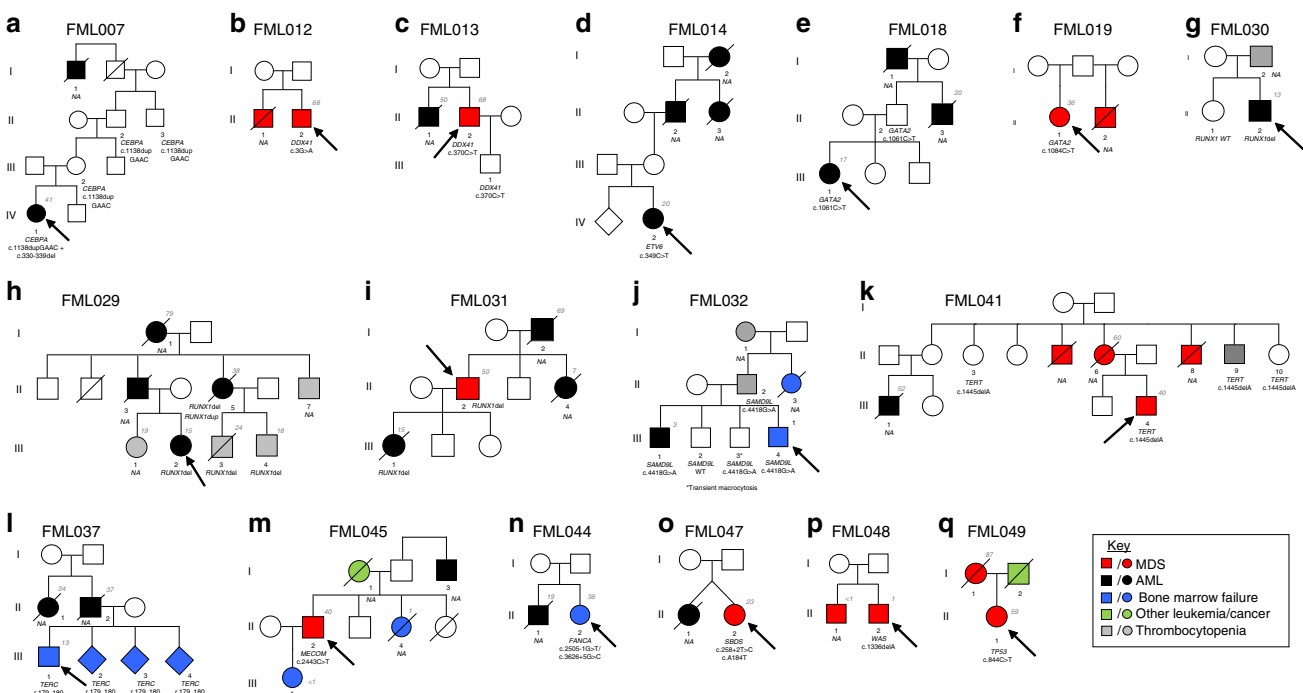

**Fig. 2 Unreported families with variants in established loci.** Pedigree representation of the 17 newly described families that include one additional *CEBPA* family (c.1138_988dupGAAC:p.Gln330Argfs*74) (**a**), two *DDX41* (c.3G>A:p.Met1? and c.370C>T:p.Arg124X) (**b, c**), one *ETV6* (c.349C>T:p.Leu117Phe) (**d**), two *GATA2* (c.1061C>T:p.Thr354Met and c.1084C>T:p.Arg362X) (**e, f**), three *RUNX1* (*RUNX1* deletions: del(21):36349450–36572837, del (21):36400658-36972948, and del(21):36389492-37056053) (**g–i**), one *SAMD9L* (c.4418G>A:p.Ser1473Asn) (**j**), one *TERT* (c.1445delA:p. His482Profs*27) (**k**), one *TERC* (r.179_180TCdelinsGG) (**l**), one *MECOM* (c.2443C>T:p.Arg815Trp) (**m**), one *FANCA* (c.2505-1G>T/c.3626+5G>C) (**n**), one *SBDS* (c.258+2T>C/c.183_184delinsCT:p.Lys62X) (**o**), one *WAS* (c.1336delA:p.Lys446fs) (**p**), and one *TP53* (c.844C>T:p.Arg282Trp) (**q**). Age of the patients is indicated as a number in gray font color and italic. Index case of each family is indicated with an arrow.

**Table 3 Genes mutated in the 17 unreported Group 1 families.**

| Family | Gene | Variant | VAF | Transcript accession number | ExAC frequency | ACMG status | Associated non-hematological features |
|---|---|---|---|---|---|---|---|
| FML007 | CEBPA | CEBPA: c.985_988dupGAAC: p.Gln330Argfs$^a$74 | NA | ENST00000498907 | No/No | Pathogenic | No |
| FML012 | DDX41 | DDX41: c.3G>A:p.Met1? | 0.454 | ENST00000507955 | 5.11E-05 | Likely pathogenic | No |
| FML013 | DDX41 | DDX41: c.370C>T:p.Arg124X | 0.61 | ENST00000507955 | No | Likely pathogenic | No |
| FML014 | ETV6 | ETV6: c.349C>T:p.Leu117Phe | 0.48 | ENST00000396373 | 8.24E-06 | VUS | No |
| FML018 | GATA2 | GATA2: c.1061C>T:p.Thr354Met (het) | 0.62 | ENST00000341105 | No | Pathogenic | No |
| FML019 | GATA2 | GATA2: c.1084C>T:p.Arg362X | 0.41 | ENST00000341105 | No | Likely pathogenic | Congenital deafness, vulva in situ neoplasia |
| FML029 | RUNX1 del | RUNX1 deletion chr21:36349450-36572837 | NA | ENST00000300305 | No | Likely pathogenic | No |
| FML030 | RUNX1 del | RUNX1 deletion chr21:36400658-36972948 | NA | ENST00000300305 | No | Likely pathogenic | No |
| FML031 | RUNX1 del | RUNX1 deletion chr21:36389492-37056053 | NA | ENST00000300305 | No | Likely pathogenic | No |
| FML032 | SAMD9L | SAMD9L: c.4418G>A:p.Ser1473Asn | 0.48 | ENST00000318238 | No | VUS | No |
| FML037 | TERC | TERC: r.179_180TCdelinsGG | NA | ENST00000602385 | No | Likely pathogenic | Prematurely gray |
| FML041 | TERT | TERT: c.1445delA:p.His482Profs$^a$27 | 0.49 | ENST00000310581 | No | Pathogenic | Skin pigmentation abnormalities |
| FML044 | FANCA | FANCA: c.2505-1G>T | 0.45 | ENST00000389301 | No | Pathogenic | Short stature, leucoplakia |
| FML045 | FANCA | c.3626+5G>C | 0.51 | ENST00000389301 | 0.000008961 | VUS | No |
| | MECOM | MECOM: c.2443C>T; p.Arg815Trp | NA | ENST00000264674 | No | Likely pathogenic | Radioulnar synostosis |
| FML047 | SBDS | SBDS: c.258+2T>C | 0.54 | ENST00000246868 | 0.003946 | Pathogenic | Learning difficulties, short stature, dysmorphic facies |
| FML049 | SBDS | c. 183_184delinsCT; p.Lys62X | 0.51 | ENST00000246868 | 0.0004118 | Pathogenic | No |
| | TP53 | TP53:c.844T:p.Arg282Trp | 0.96 | ENST00000376701 | No | Likely pathogenic | No |
| FML048 | WAS | WAS: c.133delA; p.Lys446fs | 1 | ENST00000376701 | No | Likely pathogenic | No |

aBoth FML037 and FML041 cases had short telomeres, as determined by monochrome multiplex quantitative PCR. Telomere/single-gene ratios were 0.70 and 0.54, respectively, both of which lie below the 10th centile of the normal range.

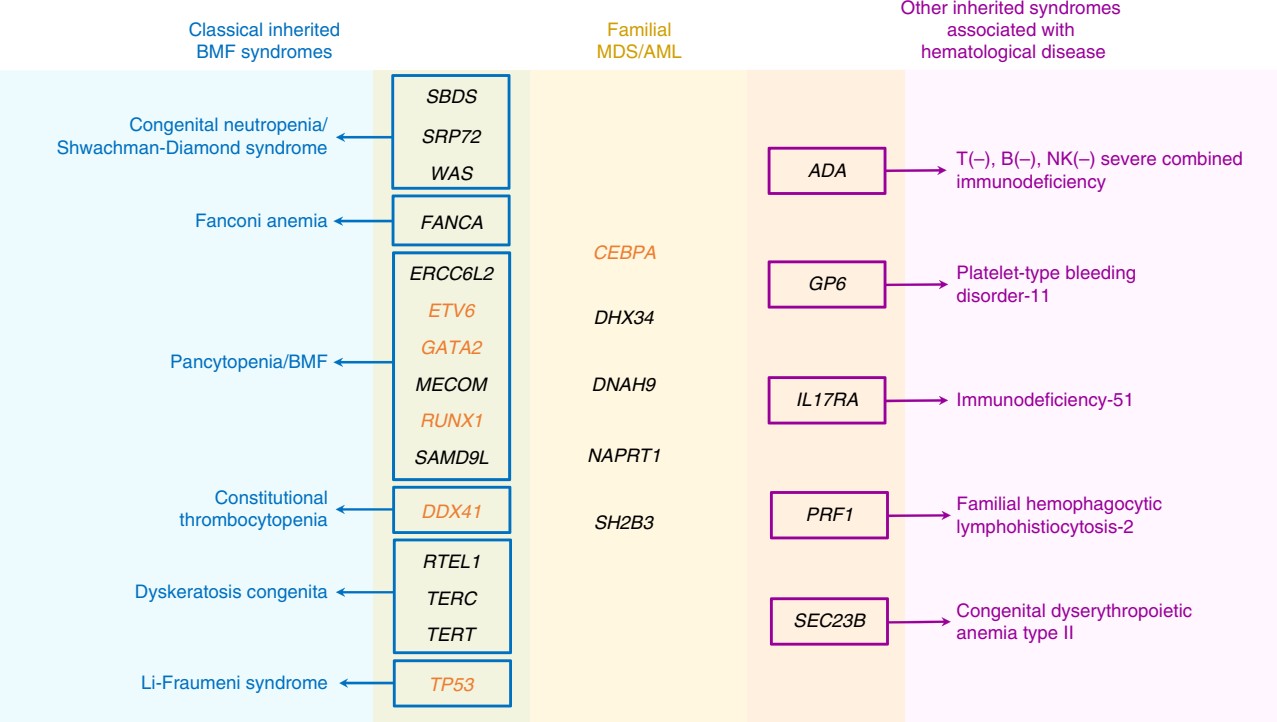

**Fig. 3 Summary of disease genes and new candidate genes mutated in our cohort of MDS/AML families.** Schematic representation of established disease genes (Group 1) and new candidate genes (Group 2) that are mutated in our cohort of MDS/AML families, and their overlap with classical inherited BMF syndromes based on Bluteau et al. classification[42] (blue) or other inherited hematological disorders (pink). In orange font, genes reported to be frequently mutated in sporadic AML.

TCGA[43] and Tyner et al.[44] data (frequency ranged from 0.1% to 11%) (Supplementary Fig. 5; Supplementary Data 8).

Following the recent study describing a synonymous *GATA2* germline variant affecting RNA splicing in a MDS/AML family[45], for completeness, we also performed an in silico analysis of our WES results using the Annovar tool[46] and the RefSeq Database[47] identifying 39 novel splicing synonymous variants (Supplementary Data 9), none of which arose in either *GATA2* or any other Group 1 loci.

**Rationalizing candidate genes into discrete groups.** We next sought to rationalize these 65 candidate genes by dividing them into two discrete groups based on an association with rare hematological syndromes (i) and a relevant function and/or mutated in three or more families (ii) (Supplementary Fig. 4). Firstly, germline variants in 26 of the selected 65 genes (40%) have been implicated across a broad range of inherited disorders (Supplementary Data 8). Notably, five genes with germline variants in 10 families, *ADA, GP6, IL17RA, PRF1,* and *SEC23B,* have been linked with autosomal recessive hematological syndromes (Fig. 3): T cell-negative, B cell-negative, natural killer cell-negative severe combined immunodeficiency (*ADA,* with novel heterozygous loss of function variants in our families FML056 (p.Gln3*) and FML071 (p.Glu88Argfs*34)); platelet-type bleeding disorder-11 (*GP6,* mutated in families FML055 (p.Arg159*) and FML063 (p.Met430Lys)); immunodeficiency-51 (IMD51) (*IL17RA,* with variants in families FML050, FML053, and FML068); familial hemophagocytic lymphohistiocytosis-2 (FHL2) (*PRF1,* mutated in family FML085 (p.Glu275Lys) and one family presented by Bluteau et al.[42] (p.Asn252Ser)); and congenital dyserythropoietic anemia type II (*SEC23B,* with variants in families FML061 (p.Pro542Leu) and FML062 (p.Glu361Asp)) (Fig. 3; Supplementary Data 8).

Among the non-syndromic genes, we noted *DNAH9* with variants in three families, a gene that interacts with the hematopoietic regulator BCL6[48] and is also mutated in one of the families described by Churpek et al.[40]. In another family we identified a variant in *SH2B3* (p.Ala49Thr), implicated in hematopoietic stem cell self-renewal and quiescence through direct interaction with JAK2[49–51], with the variant adjacent to a frameshift variant described by Bluteau et al.[42](p.Arg50-Glyfs*147) (Fig. 3). We also noted a germline duplication in *NAPRT1,* a major enzyme in cellular metabolism[52], in one additional family (FML064, p.Val216_Phe219dup), highlighting its candidacy as it was previously found to be mutated (p.Arg405*) in the affected members of an MDS/AML family described by Patnaik et al.[41] (Fig. 3; Supplementary Data 8). We also identified three families with variants in *TET2,* an epigenetic modifier that is recurrently mutated in sporadic AML and clonal hematopoiesis of indeterminate potential (CHIP), but whose germline variants may constitute a predisposing factor for myeloid neoplasm[53]. Segregation of *TET2* variant in FML054 was demonstrated by Sanger sequencing in individuals I.1 and II.2 (Supplementary Fig. 6), but we could not discount a CHIP origin of the *TET2* variants in families FML075 and FML081 (VAFs of 0.46 and 0.49, respectively).

**Variants in the nonsense-mediated RNA decay regulator *DHX34*.** Four families harbored heterozygous variants in *DHX34* (p.Tyr515Cys, p.Arg897Cys, p.Asp638Asn and p.Glu444Asp), a RNA helicase that was shown to have a functional role in the nonsense-mediated RNA decay pathway (NMD)[54,55] (Fig. 4a; Supplementary Data 8). The clinical features of the families were overlapping, with early disease onset. The index case in FML071 (II.2; Supplementary Data 3; Supplementary Fig. 2) was initially found to have thrombocytopenia, which progressed to

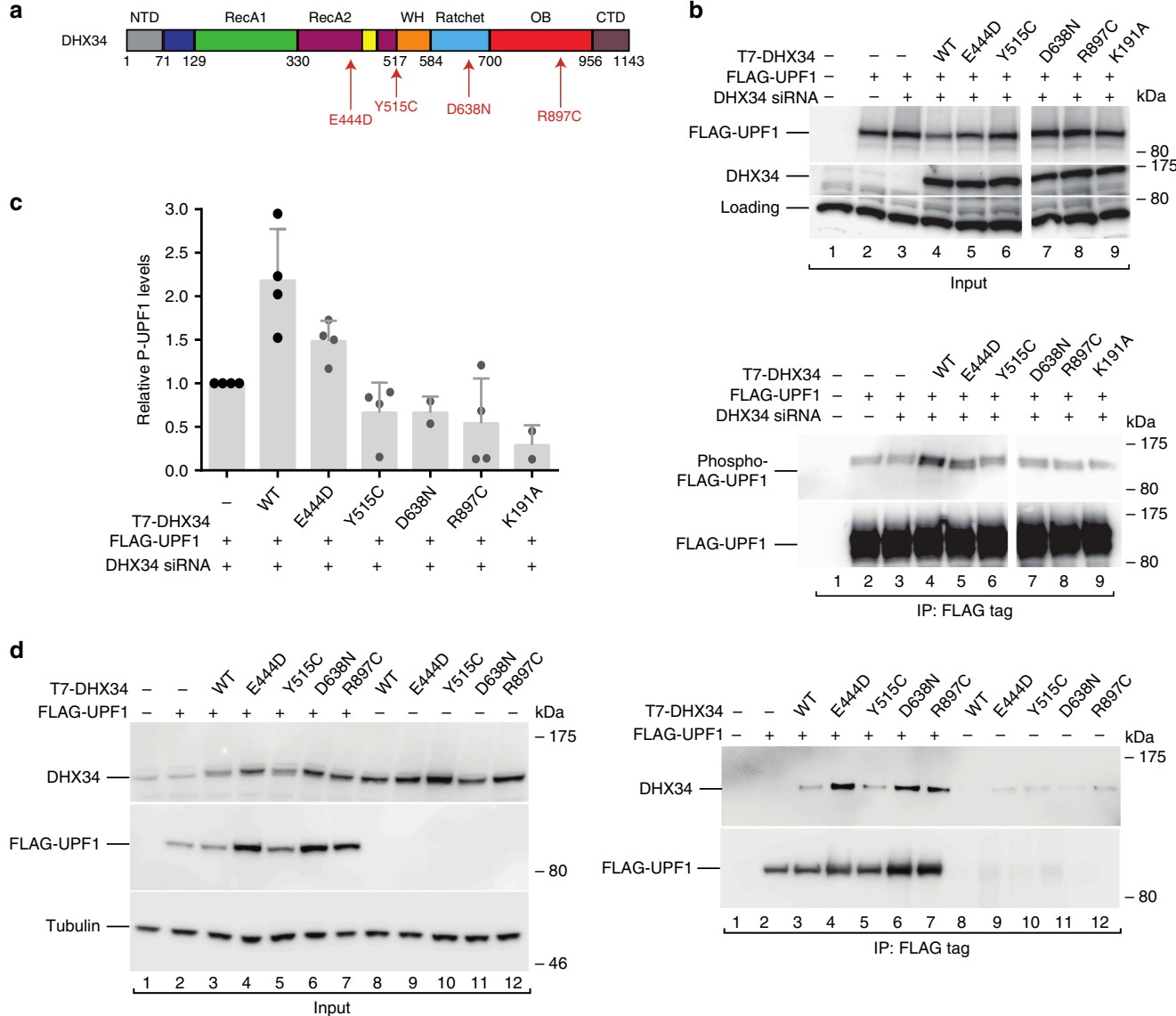

**Fig. 4 DHX34 variants fail to phosphorylate UPF1. a** Cartoon depicting the domain structure of DHX34 indicating the position of the reported variants. **b** HEK293T cells depleted of *DHX34* with a specific siRNA or transfected with a scrambled non-targeting siRNA (−) were co-transfected with FLAG-UPF1 and siRNA-resistant wild-type (WT) T7-DHX34 or the indicated *DHX34* variants (including an empty vector plasmid (−) as a control on lanes 1–2). The catalytically inactive DHX34 mutant K191A served as a negative control (lane 9). Phosphorylated UPF1 was detected with a phospho-(Ser/Thr) ATM/ATR substrate antibody and the phospho-FLAG-UPF1 signal was normalized to the levels of UPF1 recovered in the IP. Inputs (0.5%) (upper panel) and Anti-FLAG-immunoprecipitates (20%) (lower panel) were probed for the indicated proteins. **c** Quantification of the western blot signal, as shown in panel **b**. Mean values ± standard deviations from at least two independent experiments are shown. **d** Analysis of T7-DHX34 (wild-type protein (WT) and variants) binding to FLAG-UPF1. HEK293T cells were transfected with wild-type T7-DHX34 or *DHX34* variants and FLAG-UPF1. Inputs (0.5%) and Anti-T7-immunoprecipitates (20%) were probed for the presence of UPF1. Uncropped western blots are provided as a Source Data file.

pancytopenia with hypocellular BM and tri-lineage myelodysplasia at the age of 10 years. Cytogenetic analysis showed monosomy 7 in ~10% of BM cells. She died of pulmonary complications post-SCT. Her older brother (II.1) had a similar hematological profile, presenting first with thrombocytopenia and at the age of 5 years had a hypocellular BM, dysplasia, and monosomy 7. He died at the age of 11 years. In family FML061 (Supplementary Data 3; Supplementary Fig. 2), the index case (II.1) at the age of 2 years had intermittent neutropenia, anemia with macrocytosis and thrombocytopenia. The BM cellularity was reduced with tri-lineage dysplasia and monosomy 7. Her mother (I.1) was diagnosed with AML (sub-type M5) at the age of 10 years. She was treated successfully with chemotherapy. The index case in FML065 (II.1; Supplementary Data 3; Supplementary

Fig. 2) at the age of 7 years had pancytopenia and was diagnosed to have AML with normal karyotype. His father (I.2) had been diagnosed to have hypocellular myelodysplasia. In the fourth family (FML055), the index case (II.2) was found to have pancytopenia at the age of 22 years. Bone marrow examination showed he had acute mixed lineage leukemia which was refractory to chemotherapy. His older brother (II.1) was diagnosed with AML at the age of 23 years. He died from complications post-SCT. Patients with heterozygous *DHX34* variants had no extra-hematopoietic abnormalities and no acquired variants in any of the 33 genes frequently mutated in MDS/AML analyzed by targeted sequencing, apart from an *ASXL1* frameshift insertion (p. Gly646Trpfs*12) (VAF 12.7% and ExAC frequency of 0.0016) identified in the index case (II.1) of family FML061. It is

noteworthy, the index cases of two families (FML061 and FML071) had monosomy 7 and cytogenetic data were unavailable for the other two families.

Previously, we have shown that DHX34 is a bona fide NMD factor that acts in concert with core NMD factors to regulate a significant number of endogenous RNA targets in *C. elegans*, zebrafish, and human cells[56,57]. DHX34, a member of the DExH/ D box family of RNA helicases, promotes remodeling of messenger ribonucleoproteins (mRNPs) during NMD. At the molecular level, DHX34 induces a series of molecular transitions that result in an increased phosphorylation of the core NMD factor, UPF1, and leads to an enhanced recruitment of the NMD factors UPF2 and UPF3 to UPF1[55,58]. These events are required for targeting aberrant mRNAs for degradation. All four *DHX34* variants identified in our families, compromised NMD activity, as shown by their failure to promote phosphorylation of UPF1 to levels induced by the wild-type protein (Fig. 4b and c), despite retaining the ability to bind UPF1 (Fig. 4d). Importantly, we also show that all four DHX34 variants display a reduced recruitment of UPF2 and UPF3B to UPF1 (Supplementary Fig. 7).

## Discussion

While the coding landscape of somatic cases of myeloid malignancies is complete, we are still some way behind in cataloging the germline and somatic variants that arise in familial presentations of these diseases. The inclusion by the WHO of inherited forms of MDS/AML as an independent entity represented an important step forward and created a heightened awareness of this group of at risk individuals[1], so that today, with the application of NGS, more than 20 different loci of varying frequency have been linked to the occurrence of these familial diseases (Group 1; Table 1). There is a growing suspicion that the familial occurrence of AML/ MDS is underappreciated clinically[3]; however, in the absence of large collections of well annotated cases and comprehensive gene discovery programs, it will be challenging to define with confidence the precise contribution of this group of patients to MDS/ AML and to optimally manage patients and at risk family members. In our study, we have started to address this gap in knowledge by profiling the coding genome in 86 families with a history of myeloid disease. We report several new predisposing candidate genes in familial MDS/AML and note a significant overlap between the genetics of MDS/AML, inherited BMF syndromes and certain rare hematological syndromes (Fig. 3). These studies also pinpoint new biological mechanisms that appear to be specific to inherited forms of myeloid malignancies, including variants in the RNA helicase, *DHX34*, detected in four families, which impact the activity of the NMD pathway.

Overall, the pace of progress in the field of familial AML/MDS is hindered by the need for robust collection of samples, both from affected and unaffected family members and of diseased and healthy tissue, in order to discriminate germline from somatic changes, and passengers from pathogenic variants. Access to large multigenerational families is typically rare and while we have collected material on multiple affected individuals in many families, a reliance on material from a single index case remains problematic. While the availability of fibroblasts offers a gold standard approach to discriminate germline from somatic changes[17], this was not practical in our series, reflecting for the most part the historical nature of our collections. We relied on confirming the germline origin of variants by segregation analysis where material was available from multiple affected individuals. In some cases, samples were available from a single affected and unaffected family member, and while these may be informative, caution is also required since reduced penetrance or late onset of overt disease is a feature of existing Group 1 loci

that includes the *GATA2*[34], *DDX41*[59] and C-terminal *CEBPA*[33] mutations.

The candidate variants arising in our 37 discovery families exhibit a broad range of biological functions and, in some instances, have been implicated in malignant and non-malignant hematological disease (Supplementary Data 8). The confident prediction of new candidate genes is hindered by the presence of multiple variants in individual families (3.9 variants per family on average, range 1–16) and that many of the corresponding genes may not have an obvious functional association with hematopoiesis. Some candidate genes have been linked with hematological syndromes including *GP6* (FML055 and FML063), a glycoprotein receptor involved in platelet-collagen interactions during thrombus formation, mutated in platelet-type bleeding disorder-11[60]; *SEC23B* (FML061 and FML062), a component of the transport vesicle coat protein complex II (COPII), mutated in congenital dyserythropoietic anemia type II (CDAII)[61]; *ADA* (FML056 and FML071), a causative factor of T cell-negative, B cell-negative, natural killer cell-negative severe combined immunodeficiency[62]; *IL17RA* whose homozygous germline variants are the cause of Immunodeficiency-51 (IMD51); and *PRF1* (FML085) associated with familial hemophagocytic lymphohistiocytosis-2 (FHL2). Such variants highlight the phenotypic complexity that can accompany specific gene variants and it is noteworthy that in all of these cases the referring center did not observe any features of immunodeficiency or other gene-specific features. Overall, it was highly informative to cross-reference our series of variants with other discovery cohorts of familial MDS/AML or IBMF syndromes. Bluteau et al.[42] reported a variant in *SH2B3*, which is a recognized negative regulator of normal hematopoiesis that is expressed in HSCs and progenitors. *SH2B3* is mutated in 5–7% of myeloproliferative disorders, lymphoid leukemia and the non-malignant disease idiopathic erythrocytosis[63]. The variant in our study (FML085, p. Ala49Thr), which is adjacent to the one reported by Bluteau et al.[42], locates to the dimerization domain of this adapter protein and appears distinct from the variants reported in myeloproliferative neoplasms that predominate in the PH and SH2 domains[63].

No variants met our selection criteria in three families, FML052, 060, and 066 and these examples highlight perfectly the challenges faced by researchers in identifying predisposing lesions. It is conceivable that such families may not represent bona fide familial disease and instead result from multiple sporadic examples of disease in the same family. It is also feasible that the variants may reside outside of the coding region, as seen with *GATA2*[64–66], or reflect CNVs, as shown in examples of *RUNX1* mediated FPD[36]. Equally, we cannot rule out the possibility that a variant is private to a particular family; we would have overlooked the candidacy of *NAPRT1* in FML064 (p.Val216_Phe219dup) were it not for another variant (p.Arg405*) being recently reported in the same gene in an MDS/AML family by the group of Patnaik et al.[41]. This gene encodes an NAD salvage enzyme and is reminiscent of some of the recent Group 1 loci, including *DDX41*[16], as having no immediate functional association to AML/MDS. Such examples highlight the benefits of pooling data across series of patients and the importance of international collaborations, in order to focus attention on specific variants that co-occur across families, which may in the longer term gain acceptance as a Group 1 locus, while the validation of rare or private gene variation will always be challenging.

We functionally validated heterozygous variants in the RNA helicase *DHX34*, detected in four families, all of which impacted activity of the NMD (nonsense-mediated mRNA decay) pathway.

NMD is a quality-control mechanism that targets mRNAs that contain premature termination codons for degradation, but also regulates the expression levels of a large number of naturally occurring transcripts[67]. Phosphorylation of the core NMD factor UPF1 is essential for NMD activation, since this event represses further translation initiation and triggers mRNA decay[68]. Importantly, we show that all four DHX34 variants have a reduced effect on UPF1 phosphorylation (Fig. 4) and also display diminished recruitment of UPF2, indicating a decreased NMD activity (Supplementary Fig. 7). Interestingly, another RNA helicase, DDX41, is now a recognized locus in familial and acquired cases of MDS and AML[16]. DHX34 does, however, belong to a different family of RNA helicases, the DExH/D type of proteins[69], and besides its reported role in NMD, has also been identified as a component of the spliceosomal complex C, suggesting its involvement in pre-mRNA splicing[70]. The reported variants in DDX41 also give rise to defects in pre-mRNA splicing and, given the prevalence of mutations in pre-mRNA splicing factors in MDS/ AML patients[71], one possibility is that splicing defects caused by mutations in DHX34 may be one way that these germline variants contribute to pathogenesis.

While this new knowledge may well prove important for the respective families where awareness may prevent the occurrence of donor-derived disease following selection of asymptomatic donors and improve the overall management of this patient group, it is important not to overemphasize the individual impact of variants reported in our study. In a broader scheme, resolving germline predisposition can offer new insights into the etiology of acute leukemia, seen recently with DDX41[16] and SAMD9L[72,73] variants, and shed light on the mechanisms that underpin disease latency, penetrance, and intra-clinical heterogeneity, given that the disease arises from the same initiating gene mutation. We are some way off from defining the actual frequency of familial AML/ MDS and in the future, large studies like the HARMONY Alliance programme (https://www.harmony-alliance.eu/), a European Network of Excellence that is collecting information on seven hematological malignancies including MDS and AML, may provide the resources necessary to rationalize patients into distinct familial and sporadic groups and address the question of the incidence of these forms of AML/MDS. Altogether, these new data represent an important step forward in developing tailored diagnosis for familial MDS/AML while providing insights into the etiology of myeloid malignancies and a framework on which to integrate and prioritize genetic variants emerging from research and predisposing to these diseases.

## Methods

**Patients and families**. Two hundred and twenty-one individuals from 86 families were included in the study (Supplementary Data 1 and 2; Supplementary Figs. 1 and 2). In each family, two or more family members (usually first-degree relatives) were diagnosed with a hematological disorder (predominantly AML, MDS, aplastic anemia, or thrombocytopenia) with at least one of the affected cases being categorized as MDS or AML. Patients and asymptomatic members of the families included in the study gave written consent under the approval of our local research ethics committee (London—City and East, REC reference 07/Q0603/5). Exome sequencing and candidate gene sequencing were performed using genomic DNA obtained from whole peripheral blood, bone marrow aspirates, saliva or skin fibroblasts (Supplementary Data 1).

**Targeted sequencing**. Targeted sequencing of 10 known disease genes (ACD, ANKRD26, CEBPA, DDX41, ETV6, GATA2, RUNX1, SRP72, TERC, and TERT) was performed for the index case of each family using the accredited testing facility at the West Midlands Regional Genetics Laboratories (WMRGL) in Birmingham, UK [https://www.geneadviser.com/genetictest/familial_mds_familial_aml_sequencing_birmingham]. Variants were considered pathogenic if they had been previously described to be associated with familial MDS/AML or new variants (SNVs or indels) meeting the following criteria: (1) minor allele frequency in healthy population (ExAC and gnomAD databases) <0.00001, (2) variant allele frequency >30%, (3) predicted to be pathogenic with two out of four tools for

functional annotation (PolyPhen2, MutationTaster, SIFT and Provean), (4) sufficient evidence of pathogenicity and not meeting any criteria for a benign assertion following ACMG guidelines[74], and (5) where practical, the segregation with affected individuals.

A targeted myeloid panel of 33 genes frequently mutated in MDS/AML (ASXL1 (exon 12), BCOR (all), CALR (exon 9), CBL (exons 7 + 8 + 9), CEBPA (all), CSF3R (exons 14–17), DNMT3A (all), ETV6 (all), EZH2 (all), FLT3 (exons 14 + 15 + 20), GATA2 (all), GNAS (exons 8 + 9), IDH1 (exon 4), IDH2 (exon 4), IKZF1 (all), JAK2 (exons 12 + 14), KIT (exons 2, 8–11, 13 + 17), KRAS (exons 2 + 3), MPL (exon 10), NPM1 (exon 12), NRAS (exons 2 + 3), PDGFRA (exons 12, 14, 18), PHF6 (all), PTPN11 (exons 3 + 13), RUNX1 (all), SETBP1 (exon 4), SF3B1 (exons 12 – 16), SRSF2 (exon 1), TET2 (all), TP53 (all), U2AF1 (exons 2 + 6), WT1 (exons 7 + 9) and ZRSR2 (all)) was employed to determine the acquired variants in the five patients with germline DHX34 variants. An in-house True SeqCustom Amplicon (TSCA) design (Illumina, San Diego, California, USA) was used for target enrichment. Pooled library targets were sequenced the MiSeq sequencing platform. Minimum read depth threshold was 150 reads and lower limit of sensitivity was 5–10% variant allele frequency (VAF).

**Whole-exome sequencing**. WES libraries were prepared from 200 ng of genomic DNA using the Agilent SureSelect[XT] Target Enrichment System for Illumina Paired-End Multiplexed Sequencing Library coupled with the Agilent SureSelect [XT] Human all exon v6 capture reagent. Libraries were sequenced on a NextSeq 550 sequencer using the high output 300 cycles kit generating 150 bp paired-end single-indexed reads.

WES data were processed according to the pipeline described in Pontikos et al.[75] available at https://github.com/UCLGeneticsInstitute/DNASeq_pipeline. In brief, short read data were aligned using Novoalign (version 3.02.08). Variants and indels were called according to GATK (version 3.5) best practice[76]. Joint calling and variant quality score recalibration was performed with a set of 2,500 WES internal control samples from the UCLex consortium in order to minimize platform specific variants and artefactual batch effects. The variants were then annotated using the Variant Effect Predictor[77], output to JSON format, postprocessed by a Python script and loaded into a Mongo database. The filtering was performed using a bespoke R script to filter out variants. An external allele frequency threshold of less than 0.005 or missing from ExAC and GnomAD was applied. Variants with a sequencing depth <20 or in segmental duplicated regions of the genome were also filtered out.

**Sanger sequencing**. Forty-eight selected variants identified by WES in 23 candidate genes were validated by Sanger sequencing (Supplementary Fig. 6). PCR reactions to amplify each variant were performed using 1.1× ReddyMix™ (Life Technologies) and purified PCR fragments (QIAquick Gel Extraction Kit; Qiagen, Venlo, Netherlands) were sequenced by GATC Biotech (Eurofins, Ebersberg, Germany). Sequences of the oligonucleotides used are shown in Supplementary Data 10.

**RUNX1 deletions detection: array CGH, SNP array and MLPA**. As our targeted sequencing panel did not capture structural abnormalities, a combination of comparative genomic hybridization array (Array CGH) or SNP array and Multiplex Ligation-dependent Probe Amplification (MLPA) was used to identify germline RUNX1 deletions in the three families with clinical characteristics of familial platelet disorder with predisposition to myeloid malignancy (FPDMM) from our cohort (FML029, FML030 and FML031). A customized 2×400k oligoarray, Agilent™ was used in families FML029 (II.5) and FML031 (II.2) to analyze genes where CNAs have been associated with familial MDS/AML and IBMF syndromes (RUNX1, GATA2, CEBPA, SRP9, SRP14, SRP19, SRP54, SRP68, SRP72, MPL, DKC1, TERT, TERC, NOP10, NHP2, TINF2, CTC1, C16ORF57, TCAB1). DNA sample from II.2 of family FML030 was hybridized to the The Affymetrix Cytoscan HD array according to the manufacturer's protocols. Copy number analysis was performed using Affymetrix chromosome analysis suite (CHAS) v 2.1. An MLPA assay customized for familial MDS/AML loci (P437-A1 Probemix, MRC Holland) was used for verification and segregation analysis (where possible) of the novel RUNX1 deletions identified in the three families. Raw data obtained from each sample were analyzed using Peak Scanner software v1.0 (Applied Biosystems, CA) (Supplementary Fig. 3).

**Telomere length measurement**. Telomere length measurement was performed using a monochrome multiplex quantitative PCR method[78]. Briefly, the amount of telomeric DNA (T) and the amount of a single copy gene (S) were quantified in each genomic DNA sample, using standard curves in a real-time PCR reaction (Roche LightCycler). This gave a T/S ratio, which is proportional to the telomere length. Samples were run in quadruplicate and compared to the range of T/S ratios obtained from 225 healthy controls.

**DHX34 functional validation**. The plasmids pcG T7-DHX34, pcDNA3-3XFLAG-UPF1 constructs and siRNA have been described previously[55]. The DHX34 variants were cloned by PCR amplification, using full-length DHX34, as a template. Primer sequences are available upon request. For UPF1 phosphorylation and UPF2 and UPF3 recruitment experiments, HEK293T cells grown in six-well plates were

first transfected with 25 nM of siRNA using Dharmafect I (Life Technologies) following the manufacturer's instructions and expanded 24 h later. After 72 h cells were subsequently transfected with 75 nM siRNA, together with 20 µg of pcDNA3xFLAG-UPF1, and 20 µg of a plasmid expressing either siRNA-resistant wild-type or mutant T7-DHX34 protein, also including empty vector controls. Cells were harvested 48 h later for FLAG-Immunoprecipitation using Anti-FLAG M2 Affinity Gel (A2220; Sigma-Aldrich). Protein complexes were analyzed by western blotting and UPF1 phosphorylation was detected using Phospho-(Ser/Thr) ATM/ATR Substrate (2851; Cell Signaling) antibodies. FLAG-UPF1 levels in the immunoprecipitations were detected with anti-FLAG (F1804, M2 clone; Sigma-Aldrich), and further probed with anti-UPF2 (sc-20227; Santa Cruz) and anti-UPF3B (sc-48800; Santa Cruz) antibodies, as previously described[55]. Signals were detected with the ImageQuant LAS 4000 system (GE Healthcare) and quantified using the ImageQuant TL Software (GE Healthcare). For FLAG-UPF1 and T7-DHX34 co-Immunoprecipitations, cells grown in six-well plates were transfected with 1 µg pcIneo-FLAG-UPF1 and 1 µg T7-DHX34 constructs or the corresponding empty vector plasmids. Cells were expanded 24 h after and harvested 48 h after transfection. FLAG-UPF1 and T7-DHX34 constructs were detected using anti-FLAG or anti-DHX34 antibodies (non-commercial antibody described previously by Hug and Caceres)[55]. All antibodies were used at 1/1000 dilution.

## Data availability

The whole-exome sequencing data have been deposited in the EGA database under the accession code EGAS00001003399. Other datasets referenced during the study are available from TCGA[43] [https://portal.gdc.cancer.gov/projects/TCGA-LAML] and BeatAML from Tyner et al.[44] [http://www.vizome.org/aml/]. The reminder, supporting the findings of this study, are available within the article and its Supplementary Information files with information corresponding to individual samples accessible through the corresponding author, upon reasonable request. Images of uncropped western blots underlying Fig. 4b and 4d and Supplementary Fig. 7 are provided as a Source Data File. A reporting summary for this article is available as a Supplementary Information file.

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

## Acknowledgements

The authors thank the patients and their families for donating specimens for research in this study; Doriana Di Bella for her support in experimental procedures; Jessica Okosun for her critical reading of this manuscript; Sarah Charrot for her help in editing figures; and Tanya Cranfield, Charles Crawley, Raviv Dror, Ravi Kannan, Valarie Mialou, Corinne Pondarre, Elene Psiachou-Leonard, Graham R Standen, Angela Thomas, and Lisa Wolger for their contribution with patient samples. This study was predominantly funded by Bloodwise (14032), Medical Research Council (MR/PO18440/1) and Cancer Research UK (CR-UK) through a Clinical Research Fellowship awarded to K.T. N.H. and J.F.C. were supported by Core funding to the MRC Human Genetics Unit from the Medical Research Council.

## Author contributions

I.D., T.V., and J.F. conceived the project. J.F., A.R.-M., I.D., and T.V. designed the project and wrote the manuscript. A.R.-M., K.T., S.C., A.E., A.W., and H.T. performed research and collected data. A.F.A.S., J.A., and H.A. contributed to data collection. N.H. and J.F.C. designed and performed the experiments for *DHX34* functional validation. J.M., P.P., S.E.A., and K.R. performed targeted sequencing. N.P., V.P., J.W., A.R.-M., T.V., and F.C. performed data analysis. M.B., A.B., C.B., D.B., P.F., A.G., A.H., H.H.-H., U.H., S.K., S.L., M.L., A.M.M., J.M., P.M., R.M., J.F.N., C.O., J.P., E.P., C.P., R.P., N.P.-M., A.R., N.R., A.S., G.S., D.T., C.T., A.U., P.V., B.S., T.R., D.S., J.W., and J.D.C. contributed with patient samples.

## Competing interests

The authors declare no competing interests.

## Additional information

Ana Rio-Machin[1✉], Tom Vulliamy[2✉], Nele Hug[3], Amanda Walne[2], Kiran Tawana[4], Shirleny Cardoso[2], Alicia Ellison[2], Nikolas Pontikos[2], Jun Wang[5], Hemanth Tummala[2], Ahad Fahad H. Al Seraihi[1], Jenna Alnajar[2], Findlay Bewicke-Copley[1], Hannah Armes[1], Michael Barnett[6], Adrian Bloor[7], Csaba Bödör[8], David Bowen[9], Pierre Fenaux[10], Andrew Green[11], Andrew Hallahan[12], Henrik Hjorth-Hansen[13], Upal Hossain[14], Sally Killick[15], Sarah Lawson[16], Mark Layton[17], Alison M. Male[18], Judith Marsh[19], Priyanka Mehta[20], Rogier Mous[21], Josep F. Nomdedéu[22], Carolyn Owen[23], Jiri Pavlu[17], Elspeth M. Payne[24], Rachel E. Protheroe[20], Claude Preudhomme[25,26], Nuria Pujol-Moix[22], Aline Renneville[27], Nigel Russell[28], Anand Saggar[29], Gabriela Sciuccati[30], David Taussig[31], Cynthia L. Toze[6], Anne Uyttebroeck[32], Peter Vandenberghe[32], Brigitte Schlegelberger[33], Tim Ripperger[33], Doris Steinemann[33], John Wu[34], Joanne Mason[35], Paula Page[35], Susanna Akiki[36], Kim Reay[35], Jamie D. Cavenagh[37], Vincent Plagnol[38], Javier F. Caceres[3], Jude Fitzgibbon[1,40✉] & Inderjeet Dokal[2,39,40✉]

[1]Centre for Haemato-Oncology, Barts Cancer Institute, Queen Mary University of London, London, UK. [2]Centre for Genomics and Child Health, Blizard Institute, Queen Mary University of London, London, UK. [3]MRC Human Genetics Unit, Institute of Genetics and Molecular Medicine, University of Edinburgh, Edinburgh, UK. [4]Department of Haematology, Addenbrooke's Hospital, Cambridge, UK. [5]Centre for Molecular Oncology, Barts Cancer Institute, Queen Mary University of London, London, UK. [6]The Leukemia/BMT Program of British Columbia, Division of Hematology, Department of Medicine, Faculty of Medicine, University of British Columbia, Vancouver, BC, Canada. [7]Department of Haematology, Christie Hospital, Manchester, UK. [8]MTA-SE Lendulet Molecular Oncohematology Research Group, 1st Department of Pathology and Experimental Cancer Research, Semmelweis University, Budapest, Hungary. [9]Department of Haematology, St James's University Hospital, Leeds, UK. [10]Service d'hématologie Séniors, Hôpital St Louis/Université Paris, Paris, France. [11]National Centre for Medical Genetics, Our Lady's Children's Hospital, Crumlin, Dublin, Ireland. [12]Children's Health Queensland Hospital and Health Service, Queensland Children's Hospital, South Brisbane, QLD, Australia. [13]Department of Hematology, St Olavs Hospital and Institute of Cancer Research and Molecular Medicine (IKM) Norwegian University of Science and Technology (NTNU), Trondheim, Norway. [14]Department of Haematology, Whipps Cross Hospital, Barts NHS Trust, London, UK. [15]Department of Haematology, The Royal Bournemouth Hospital NHS Foundation Trust, Bournemouth, UK. [16]Department of Haematology, Birmingham Children's Hospital, Birmingham, UK. [17]Centre for Haematology, Imperial College London, Hammersmith Hospital, London, UK. [18]Clinic Genetics Unit, Great Ormond Street Hospital, London, UK. [19]Department of Haematological Medicine, Haematology Institute, King's College Hospital, London, UK. [20]Bristol Haematology Unit, University Hospitals Bristol NHS Foundation Trust, Bristol, UK. [21]UMC Utrecht Cancer Center, Universitair Medisch Centrum Utrecht, Huispostnummer, Utrecht, Netherlands. [22]Laboratori d´Hematologia, Hospital de la Santa Creu i Sant Pau, Universitat Autònoma de Barcelona, Barcelona, Spain. [23]Division of Hematology and Hematological Malignancies, Foothills Medical Centre, Calgary, AB, Canada. [24]Department of Haematology, UCL Cancer Institute, University College London, London, UK. [25]Laboratory of Hematology, Biology and Pathology Center, Centre Hospitalier Regional Universitaire de Lille, Lille, France. [26]Jean-Pierre Aubert Research Center, INSERM, Universitaire de Lille, Lille, France. [27]Broad Institute of Harvard and MIT, Cambridge, MA, USA. [28]Centre for Clinical Haematology, Nottingham University Hospitals NHS Trust, Nottingham, UK. [29]Clinical Genetics, St George's Hospital Medical School, London, UK. [30]Servicio de Hematologia y Oncologia, Hospital de Pediatría "Prof. Dr. Juan P. Garrahan", Ciudad Autonoma de Buenos Aires, Argentina. [31]Haemato-oncology Department, Royal Marsden Hospital, Sutton, UK. [32]Department of Hematology, University Hospitals Leuven, Leuven, Belgium. [33]Institut für Humangenetik, Medizinische Hochschule Hannover, Hannover, Germany. [34]British Columbia Children's Hospital, Vancouver, BC, Canada. [35]West Midlands Regional Genetics Laboratory, Birmingham Women's NHS Foundation Trust, Birmingham, UK. [36]Department of Laboratory Medicine & Pathology, Qatar Rehabilitation Institute, Hamad Bin Khalifa Medical City (HBKM), Doha, Qatar. [37]Department of Haematology, St Bartholomew's Hospital, Barts NHS Trust, London, UK. [38]Genetics Institute, University College London, London, UK. [39]Barts Health NHS Trust, London, UK. [40]These authors jointly supervised this work: Jude Fitzgibbon, Inderjeet Dokal. ✉email: a.rio-machin@qmul.ac.uk; t.vulliamy@qmul.ac.uk; j.fitzgibbon@qmul.ac.uk; i.dokal@qmul.ac.uk

