## [Peer Review File · Nature Communications]

Reviewers' comments:

Reviewer #1 (Remarks to the Author):

The report by Rio-Manchin et al describes the genetic findings of 86 families that have multiple members affected by a hematologic malignancy. They report that over half of the families have a mutation in one of 15 previously reported genes. In addition, WES identified in the remaining cases identified multiple putative novel causes of familial MDS/AML. This is an interesting study that will make an impact on the field. There are a number of concerns, both major and minor, that require attention.

1. More information in the main text needs to be included to describe the overall sequencing workflow. My impression is that all patients first got panel sequencing and then those without a mutation in genes present on that panel were characterized by WES. Is that accurate?
2. In the supplemental methods the authors describe a panel with 10 group 1 genes: ACD, ANKRD26, CEBPA, DDX41, ETV6, GATA2, RUNX1, SRP72, TERC and TERT. SAMD9 and SAMD9L are not list here but are included in Group 1 in Table 1. What sequencing approach was used for SAMD9 or SAMD9L?
3. It appears that BM or PB was used for all samples. Did the authors confirm germline status for any of these variants? Bone marrow fibroblasts or sorted T cells could be used buccal/skin samples are not available.
4. Why is TP53 not listed in Table 1?
5. The authors state in the first sentence after the summary that "Inherited forms of MDS/AML probably account for <5% of new diagnoses of all myeloid malignancies". Please provide justification for this statement, preferably with a reference.
6. Please provide more information for how the 86 families were identified and collected. From a national or local database? How many were originally identified but didn't have available samples?
7. The sentence "In this study, familial AML/MDS references families where two or more family members were diagnosed with a hematological malignancy (predominantly AML, MDS, aplastic anemia) or a prodromal cytopenia (e.g. thrombocytopenia) but at least one case was MDS or AML" is confusing and not a complete sentence.
8. Coincide, not "co-inside".
9. Were telomere length studies, which are fairly standard, performed on FML037 with the TERC variant?
10. This sentence is unclear: "To mitigate the large number of missense variants, we excluded missense variants in the same gene arising in only 2 families.." Do the authors mean in less than 2 families?
11. Please include CADD and REVEL scores for all variants and include them in Table 3.
12. The authors describe novel variants in ADA, GP6 and SEC23B in different families. Did any of these families have features associated with these genes? For example, did families FML056 and FML071 have features of immunodeficiency?
13. In all the pedigrees, does the arrow reflect the patient whose sample was sequenced?
14. Figure 1: SAMD9L is listed as group 2 but it is group 1 in Table 1.
15. The authors state that DHX34 promotes NMD and that all 4 variants "compromised NMD activity". This is presumably based on the decreased phosphorylation of UPF1. Do the authors have any other data to support that NMD is affected?
16. Please include full westerns for the blots in Figure 4. What is the "loading" control in panel B?
17. The data in panel d of Figure 4 is not convincing that wild-type or mutants (with the exception of R987C) bind UPF1. I see some FLAG-UPF1 in the control lane #2 (no T7-DHX34).
18. The authors are detecting phospho-UPF1 using an ATM/ATR substrate antibody after anti-FLAG IP. Is there evidence that the band shown in the bottom part of panel B labeled "phosphor-FLAG-UPF1" is indeed UPF1 rather than another phosphorylated protein that is pulled down in a complex with UPF1?

Reviewer #2 (Remarks to the Author):

In this article, Rio-Machin et al use targeted panel-based sequencing and whole exome sequencing to investigate 86 families with two or more cases of MDS/AML/MPN or thrombocytopenia. Of these, 32 families have already been published but are included here for description of spectrum of involved genes and mutations; Of the remaining unpublished families, 16 new families with variants in known familial MDS/AML or marrow failure genes are reported and candidate genes identified by exome sequencing in the 38 remaining unsolved families are reported. The study investigates an important question in a valuable series of familial cases, but could be improved by the following:

Major comments

1. The most major issue comes with tissue sample used for detecting germline mutations. The authors used blood or marrow samples from individuals affected with MDS/AML. Thus, for families with proband only sequencing (e.g. pedigree D in Figure 1; unknown number of pedigrees in supplement), how can one be sure that the findings are germline and not acquired? This limits conclusions drawn from the data as currently presented. To strengthen their case, the authors should include:

A. Known syndrome cases: Table 3 should include VAF of variants, tissue source, disease status at time of blood/marrow sample used for sequencing, cis/trans status of those with two mutations (e.g. the SBDS case) and any other corroborating evidence to support that the patient has the syndrome (e.g. chromosome breakage testing in the FANCA case) or clinical features known to be associated with the syndrome (more ideal in main table than in supplement)

B. Supplemental Figure 3 pedigrees: indicate which individuals in each family were sequenced and samples used; given overlap of some of the genes identified and known acquired changes in these genes occurring in MDS/AML/marrow failure (23% of the identified variants per page 2), how can one be sure that the variants described are not just recurring acquired mutations? TET2 and TP53, for example, were only examined in one individual per family and could very likely be acquired with high VAF. There is already evidence that individuals with a germline gene mutation in RUNX1 and MBD4, are more likely to have CHIP so these familial cases may have higher prevalence of CHIP and further confounds data from peripheral blood and marrow sequencing to find germline events.

2. Categorization of known syndrome versus novel associations. At present, the table 1 categories (group 1 and group 2) as well as those presented in the venn diagram for figure 3 require further refinement. It is confusing to have genes like TP53 described in the novel associations category when there are published cases of familial leukemia with this gene. Further, group 1 and 2 seem arbitrary per comments for this table below. Given the tissue source issue as described above, more caveats are needed in the text and in presenting the novel gene association data to make it clear that some of these findings could be acquired. With that in mind, reconsider Figure 3 presentation

Minor comments:

1. Page 1: the first sentence of the main text (inherited forms of ...) is speculative and lessens the reader's enthusiasm for the data that follows

2. Page 1: Definition of familial AML/MDS should be more precise:

What degree of relatives (first, second, third, more?)

Define length of prodromal cytopenia (are these cytopenias preceding a hem malignancy diagnosis or persistent without transformation)

3. Page 1: incomplete penetrance comment: if NA signifies DNA not available (please clarify this in the Key or in the figure legend), then FML014 did not include genotyping of additional family members to confirm germline status and/or segregation. FML017 does not appear to have incomplete penetrance (and this pedigree is in supplement); FML041 does not have genotyping to confirm incomplete penetrance in the proband's aunt

4. Page 1: other non-hematological features comment: these features should be added to Table 3 as described above to convince the reader that patient had the associated syndrome
5. Page 2: "where genetic screening must venture outside the coding genome": Confusing as stated. Could start that sentence at the "Germline RUNX1 deletions..." statement as seems more like emphasis is on CNV analysis rather than non-coding regions
6. Given that 32 of the 48 families with a primary genetic lesion identified were already known, a comment indicating that this series may not reflect yield of testing in fully uncharacterized familial mds/aml cases should be added.

Figure 1:

1. Would be more effective if presented as number of families with variants in the genes on the x axis sufficient for a clinical diagnosis of the respective syndrome. Those with VUS or heterozygous changes in genes in which homozygosity or compound heterozygosity is more convincingly diagnostic of a syndrome should be left out or highlighted in a different way so it is easier for reader to appreciate

Figure 2:

1. pedigrees would benefit from age of onset of the hematologic malignancies
2. clarify meaning of "NA" in the key or figure legend

Figure 3:

1. TP53 germline mutations are known to contribute to leukemia risk; this gene should really be in group1/2
2. venn diagram also seems arbitrary in its divisions; how are DHX34, etc in familial mds/aml but DDX41 is considered a classical inherited BMF syndrome with constitutional thrombocytopenia? This diagram could be removed

Table 1:

1. Group 1 versus Group 2 would benefit from more refined definitions. As currently presented, having genes like FANCA and SBDS for which approved testing exists and there is a clear predisposition to MDS/AML placed into Group 2 versus SAMD9 and SAMD9L in Group 1 (testing is harder to get for these new genes) is confusing. Consider separating into:
Group 1: Families with variants diagnostic of a known familial MDS/AML or inherited marrow failure syndrome (this would then include these Fanconi anemia and SDS cases); if exome sequencing was performed on all cases, then genes included in this category should absolutely be expanded (e.g. even TP53 is not on this list and is known cause of familial mds/aml)
Group 2: Families with variants in preliminary evidence genes. This category could then specify genes for which heterozygous variants were found in genes that homozygous or compound heterozygous mutations are more strongly associated with hematologic malignancy (e.g. RTEL1 or PARN)

Table 2:

1. Define family history of hematological disease more precisely

Table 3:

1. see comments above

Reviewer #3 (Remarks to the Author):

The present study is led by a highly reputable group and focuses on the characterization of germline mutations in myeloid disease. The authors have in their hands a truly unique cohort. This study has the potential to significantly advance our understanding of the specific mutations and

genes that contribute towards inherited risk for myeloid neoplasia. These findings can shape development of clinical diagnostic assays for the detection of germline predisposition variants, inform genetic counselling practices and uncover new biology. However, despite this significant resource, the manuscript suffers from lack of methodological detail as well as structure in the presentation of findings. A significant proportion of the relevant information is dispersed amongst the figures and supplemental material. The paper in itself is hard to follow and put together. This precludes it from publication in its current form. The authors should consider investing some effort in presenting the samples, methods and notation of their findings before resubmission of this manuscript. A few recommendations for the manuscript are provided below:

The authors do not provide any detail on the results section on the number of individuals tested or ascertained. The authors do not discuss detailed methods for the annotation of each mutation. In the method's the authors mention under criterion (5) that "practical segregation amongst affected individuals was considered". However, none of this information is provided in the text. How many families had more than 1 proband tested? Were any unaffected individuals tested? The authors should provide a very detailed overview of the number of families with 1, 2, 3 or more cases available for testing in the main text as well as supplemental material. The authors should provide information on the timing of sampling acquisition for molecular profiling and how they addressed biases from sampling time relative to manifestation of clinical phenotype.

In the methods the authors say that they performed targeted gene resequencing and prepared whole exome sequencing libraries (which is valid, as this reflects the right library preparation method for capture based assays). However, the authors then proceed in discussing analyses of whole exome sequencing data. The authors should clarify the nature of the assay and data generated for this project. The authors discuss 10 genes in the methods, but show more genes in both tables and main figures. The authors should be very specific on what assay was generated for how many individuals.

The authors provide very little methodological detail on the post processing and filtering criteria of the data. How did they authors distinguish between rare CHIP/ARCH variants and rare germline events. For example, mutations in TET2 are frequently seen CHIP/ARCH as clonal with low population based minor allele estimates. The authors discuss the detection of 4 putative germline variants in TET2 as potentially new variants but do not discuss their methodology neither their validation of this observation. One validation of such events as germline would be to identify them in related and affected individuals. However, such data are referenced in the text, nor how these were specifically excluded from being CHIP variants.

Please provide a detail summary in the main text of total number of families, affected individuals and non-affected individuals (criteria for non-affected) for who you ascertained samples for this study.

When providing summary information on the germline data, please include information on whether the precise mutation identified has been previously observed and validated as germline as opposed to representing a previously unseen. Also include in supplemental material the categorization of previously known versus previously unseen mutation as mapped by 1 affected individual, 2, 3 or more.

The authors discuss functional validation but do not show any functional such data or discuss it in any detail in the text.

Did the authors have access to diagnostic samples? It would be interesting to test whether at diagnoses which of these variants often loose the wild type allele. Such an observation could serve as a further validation or evidence of the putative oncogenic role of such mutations.

Reviewer #1 (Remarks to the Author):

The report by Rio-Manchin et al describes the genetic findings of 86 families that have multiple members affected by a hematologic malignancy. They report that over half of the families have a mutation in one of 15 previously reported genes. In addition, WES identified in the remaining cases identified multiple putative novel causes of familial MDS/AML. This is an interesting study that will make an impact on the field. There are a number of concerns, both major and minor, that require attention.

1. More information in the main text needs to be included to describe the overall sequencing workflow. My impression is that all patients first got panel sequencing and then those without a mutation in genes present on that panel were characterized by WES. Is that accurate?

Your comments on the overall structure have been most helpful and the revised manuscript has improved accordingly. We agree wholeheartedly that the manuscript would benefit from significant additional information, including both the sequencing work-flow and our sample collections. These have now been included in Supplementary Table 3. The spreadsheet includes information for all 86 families (FML001-086) detailing both the number of (i) subjects (ii) samples (iii) sample type (iii) segregation data (iv) sequencing approaches and an overall summary of the collection. A description of the series as a whole and indeed the limitations inherent in such a retrospective series is also described for completeness in both the results section and throughout the entirety of the manuscript.

2. In the supplemental methods the authors describe a panel with 10 group 1 genes: ACD, ANKRD26, CEBPA, DDX41, ETV6, GATA2, RUNX1, SRP72, TERC and TERT. SAMD9 and SAMD9L are not list here but are included in Group 1 in Table 1. What sequencing approach was used for SAMD9 or SAMD9L?

The *SAMD9L* variant was identified by exome sequencing.

3. It appears that BM or PB was used for all samples. Did the authors confirm germline status for any of these variants? Bone marrow fibroblasts or sorted T cells could be used buccal/skin samples are not available.

We first of all, would agree with the reviewer that discriminating germline from acquired variants is of critical importance. The availability of fibroblasts or sorted T cells, offers a gold standard approach that was not practical, given the historical nature of our collections. That said, we had an opportunity to test many variants across >1 affected individual as well as the use of remission or buccal DNA as now detailed in Supplementary Table 3 and while not confirmatory by any means the median VAF of all 130 of our selected variants was 0.48.

4. Why is TP53 not listed in Table 1?

We have now included *TP53* in the 'known genes' Group 1 category.

5. The authors state in the first sentence after the summary that "Inherited forms of MDS/AML probably account for <5% of new diagnoses of all myeloid malignancies". Please provide justification for this statement, preferably with a reference.

The frequency of familial MDS/AML is an open question and our statement that it ‘probably account for <5% of new diagnoses’ has been removed. It is more accurate to say that familial AML/MDS ‘are thought to be rare but the precise figures on its incidence and prevalence are not known’ and we have now amended the revised manuscript to reflect our current knowledge appropriately. Reflecting on the comments of the reviewer, we have also included a section on incidence and how this may better be best addressed in the discussion, through new initiatives like the Harmony consortium.

6. Please provide more information for how the 86 families were identified and collected. From a national or local database? How many were originally identified but didn’t have available samples?

Regrettably, we do not have either a national or local database in place today. Indeed, our experience is that many hematologists are only now becoming aware of this group of patients, more so since the inclusion of familial myeloid malignancies as a separate disease entity in the World Health Organization (WHO) classification of hematological cancers. Our collections have therefore very much reflected ‘word of mouth’ or ‘following publications’ where both national and international collaborators (13 countries included in the study) have provided material for analysis. These points have now been stated clearly in the revised manuscript.

7. The sentence “In this study, familial AML/MDS references families where two or more family members were diagnosed with a hematological malignancy (predominantly AML, MDS, aplastic anemia) or a prodromal cytopenia (e.g. thrombocytopenia) but at least one case was MDS or AML” is confusing and not a complete sentence.

This sentence has been re-written, to clarify the inclusion criteria. “In this study, familial AML/MDS refers to families where two or more family members (usually first degree) were diagnosed to have a hematological disorder (predominantly AML, MDS, aplastic anemia, thrombocytopenia) with at least one of the affected cases being categorized as MDS or AML”.

8. Coincide, not “co-inside”.

Corrected

9. Were telomere length studies, which are fairly standard, performed on FML037 with the TERC variant?

Yes, we have performed telomere length analysis in this and the TERT families. A note to this effect has been appended as a footnote to Table 3 and our telomere length measurement protocol has been added to Methods section.

10. This sentence is unclear: “To mitigate the large number of missense variants, we excluded missense variants in the same gene arising in only 2 families..” Do the authors mean in less than 2 families?

This section has been rewritten in order to explain the pipeline used in rationalizing our WES into variants of interest, in particular how missense variants were treated which represented 86.3% of all the genetic lesions reported.

11. Please include CADD and REVEL scores for all variants and include them in Table 3.

This has now been done and is included in an extended version of the Table 3 in Supplementary Table 4.

12. The authors describe novel variants in ADA, GP6 and SEC23B in different families. Did any of these families have features associated with these genes? For example, did families FML056 and FML071 have features of immunodeficiency?

This is a very pertinent point and we have been in contact with the referring physicians and can state that they do not observe any features of immunodeficiency in the two *ADA* families (FML056 and FML071) or other relevant disease features in families with *GP6* (FML055 and FML063) and *SEC23B* (FML061 and FML062) variants. This information has been now added to the manuscript.

13. In all the pedigrees, does the arrow reflect the patient whose sample was sequenced?

In all families, the arrow reflects the index case, and indeed these cases were the examples taken forward for WES. Additional family members where samples were also sequenced are now indicated with an asterisk to Supplementary Figure 2.

14. Figure 1: SAMD9L is listed as group 2 but it is group 1 in Table 1.

Thank you for pointing out this inconsistency. In our revision we have now reassigned our families into 2 discrete groups, Group 1 families (FML001-049; Supplementary Figure 1; Supplementary Data 1) where a variant has been detected either in an established locus (including SAMD9L) or emerging from basic research but requiring more validation and Group 2 (Supplementary Figure 2; Supplementary Data 2), where the molecular lesion has not been defined.

15. The authors state that DHX34 promotes NMD and that all 4 variants “compromised NMD activity”. This is presumably based on the decreased phosphorylation of UPF1. Do the authors have any other data to support that NMD is affected?

We now show in Supplementary Figure 6 that all 4 *DHX34* variants display a decreased recruitment of the NMD factor UPF2 and UPF3b to UPF1. Thus, we have successfully demonstrated that the variants reported here have a decreased activity in two events that are essential for NMD activation i) UPF1 phosphorylation (Fig. 4) and ii) recruitment of UPF2 and UPF3b to UPF1 (Supplementary Figure 6). This is also now expanded upon in the Result sections where we have a separate section, describing *DHX34* families and the functional assessment of these *DHX34* variants.

16. Please include full Westerns for the blots in Figure 4. What is the “loading” control in panel B?

This is a very reasonable request and we are now including full Western blots as part of the source data. The loading control in Fig. 4 is an unspecific band that is revealed using the anti-*DHX34* antibody.

17. The data in panel d of Figure 4 is not convincing that wild-type or mutants (with the exception of R987C) bind UPF1. I see some FLAG-UPF1 in the control lane #2 (no T7-DHX34).

The referee quite correctly has referred to some background binding in the negative control, which represents an immunoprecipitation of T7-beads in cell extracts overexpressing FLAG-UPF1 and T7 empty vector. We observed this background binding in several experiments. However, what we consider to be important here, is that the co-purification of DHX34 with FLAG-beads is increased upon overexpression of FLAG-UPF1, as compared to the FLAG empty vector (control). In order to improve upon this finding, we have now repeated the experiment using FLAG-UPF1 as a bait and show that all DHX34 proteins, including WT and DHX34 variants bind to UPF1 above background (see new panel d on Fig. 4).

18. The authors are detecting phospho-UPF1 using an ATM/ATR substrate antibody after anti-FLAG IP. Is there evidence that the band shown in the bottom part of panel B labeled “phosphor-FLAG-UPF1” is indeed UPF1 rather than another phosphorylated protein that is pulled down in a complex with UPF1?

It is worth pointing out that this is a well, established assay in the NMD field. We and others have used this assay in the past and showed that mutations in UPF1 changed this phosphorylation pattern (see Hug and Caceres (2014) Cell reports | PMID: 25220460; Durand et al. (2016) Nat comm | PMID: 27511142). In agreement with a decreased phosphorylation of UPF1 in the presence of the DHX34 variants (Fig. 4b) we have also observed a reduced recruitment of UPF2 to UPF1 (see Supplementary Figure 6).

Reviewer #2 (Remarks to the Author):

In this article, Rio-Machin et al use targeted panel-based sequencing and whole exome sequencing to investigate 86 families with two or more cases of MDS/AML/MPN or thrombocytopenia. Of these, 32 families have already been published but are included here for description of spectrum of involved genes and mutations; Of the remaining unpublished families, 16 new families with variants in known familial MDS/AML or marrow failure genes are reported and candidate genes identified by exome sequencing in the 38 remaining unsolved families are reported. The study investigates an important question in a valuable series of familial cases, but could be improved by the following:

Major comments

1. The most major issue comes with tissue sample used for detecting germline mutations. The authors used blood or marrow samples from individuals affected with MDS/AML. Thus, for families with proband only sequencing (e.g. pedigree D in Figure 1; unknown number of pedigrees in supplement), how can one be sure that the findings are germline and not acquired? This limits conclusions drawn from the data as currently presented. To strengthen their case, the authors should include:

A. Known syndrome cases: Table 3 should include VAF of variants, tissue source, disease status at time of blood/marrow sample used for sequencing, cis/trans status of those with two mutations (e.g. the SBDS case) and any other corroborating evidence to support that the patient has the syndrome (e.g. chromosome breakage testing in the FANCA case) or clinical features known to be associated with the syndrome (more ideal in main table than in supplement)

As we have also reported to the other reviewers, your comments on the overall structure have been most helpful and the revised manuscript has improved accordingly. We agree wholeheartedly that the manuscript would benefit from significant additional information, particularly as noted with respect to our sample collections. These have now been detailed in a new Supplementary Table 3. The spreadsheet includes information for all 86 families (FML001-086) detailing both the number of (i) subjects (ii) samples (iii) sample type (iii) segregation data (iv) sequencing approaches and an overall summary of the collection. A description of the series as a whole and indeed the limitations inherent in such a retrospective series is also described for completeness in both the results section and throughout the manuscript.

B. Supplemental Figure 3 pedigrees: indicate which individuals in each family were sequenced and samples used; given overlap of some of the genes identified and known acquired changes in these genes occurring in MDS/AML/marrow failure (23% of the identified variants per page 2), how can one be sure that the variants described are not just recurring acquired mutations? TET2 and TP53, for example, were only examined in one individual per family and could very likely be acquired with high VAF. There is already evidence that individuals with a germline gene mutation in RUNX1 and MBD4, are more likely to have CHIP so these familial cases may have higher prevalence of CHIP and further confounds data from peripheral blood and marrow sequencing to find germline events.

We agree with the reviewer that discriminating germline from acquired and indeed pathogenic from passenger variants is of critical importance. Our collections have been accrued over a significant period of time, by ‘word of mouth’ or ‘following publications’ where both national and international collaborators (13 countries included in the study) have provided material for analysis. The availability of fibroblasts or sorted T cells, which would provide a gold standard approach to discriminate germline from acquired is not practical in our series, given the historical nature of our collections. That said, we had an opportunity to test many variants across >1 affected individuals, the use of remission or buccal DNA or lastly through extensive *in silico* analysis and document the frequency of these variants in the ExAc and gnomAD databases (Supplementary Table 3) and while not confirmatory by any means the median VAF of all 130 of our selected variants was 0.48.

Several studies documenting the *TP53* variant identified FML051 (c.C844T:p.Arg282Trp) in Li-Fraumeni and “Li-Fraumeni-Like” syndromes (Toguchida et al., 1992; Iavarone et al., 1992; Malkin et al., 1992; Smith-Sorensen et al., 1993) and the low frequency on ExAC (0.00001659), confirm that this variant is disease predisposing in our FML051 family, and is now included in our Group 1 families. In the case of *TET2* variants, we have demonstrated segregation in FML054 (p. Ala241Val) where DNA from patient I.1 was available (validation now in Supplementary Figure 5). Only the index case sample was available in the other two families (FML075, p.Cys1464*, VAF=0.46; and FML081, p.Pro1962Leu, VAF=0.41) and we accept the point raised by the reviewer that may be acquired and the need for caution. We have therefore nuanced our description of these *TET2* variants accordingly in the manuscript.

2. Categorization of known syndrome versus novel associations. At present, the table 1 categories (group 1 and group 2) as well as those presented in the venn diagram for figure 3 require further refinement. It is confusing to have genes like TP53 described in the novel association category when there are published cases of familial leukemia with this gene. Further, group 1 and 2 seem arbitrary per comments for this table below. Given the tissue source issue as described above, more caveats are needed in the text and in presenting the novel gene association data to make it clear that some of these findings could be acquired. With that in mind, reconsider Figure 3 presentation

Your comments on the overall structure have been most helpful and the revised manuscript has improved accordingly. In our resubmission we have now rationalized our families into 2 discrete groups, Group 1 families (FML001-049; Supplementary Figure 1; Supplementary Data 1) where a variant has been detected either in an established locus or emerging from basic research but requiring more validation and Group 2 (FML050-086 Supplementary Figure 2; Supplementary Data 2), where the molecular lesion has not been defined. It is sensible that *TP53* is now assigned to Group 1 as an established locus. As detailed earlier, in response to comment 1 above, we agree with the reviewer that discriminating germline from acquired and indeed pathogenic from passenger variants is of critical importance and that the confidence in each variant should not be overstated on either account. We have very consciously made it clear throughout the revised manuscript that we are herein proposing rather than confirming the existence of several new predisposing genes/variant responsible for familial MDS/AML.

Minor comments:

1. Page 1: the first sentence of the main text (inherited forms of ...) is speculative and lessens the reader's enthusiasm for the data that follows

This has now been amended.

2. Page 1: Definition of familial AML/MDS should be more precise: What degree of relatives (first, second, third, more?) Define length of prodromal cytopenia (are these cytopenias preceding a hem malignancy diagnosis or persistent without transformation)

In the second paragraph of the introduction we have modified the sentence regarding the definition of familial MDS/AML. This now reads "In this study, familial AML/MDS refers to families where two or more family members (usually first degree) were diagnosed to have a hematological disorder (predominantly AML, MDS, aplastic anemia, thrombocytopenia) with at least one of the affected cases being categorized as MDS or AML". We have also removed reference to prodromal cytopenias as this has caused confusion. We were merely referring to individuals in some families who had thrombocytopenia at the time of investigation while other first degree relatives in the same family had been diagnosed to have MDS or AML or aplastic anemia.

3. Page 1: incomplete penetrance comment: if NA signifies DNA not available (please clarify this in the Key or in the figure legend), then FML014 did not include genotyping of additional family members to confirm germline status and/or segregation. FML017 does not appear to have

incomplete penetrance (and this pedigree is in supplement); FML041 does not have genotyping to confirm incomplete penetrance in the proband's aunt

We have clarified the meaning of NA. It is true that in FML014 we only had samples from the index case. However, we have been able to test a remission sample and this demonstrates clear heterozygosity post treatment, establishing the variant is of germline origin. This information is now included in Supplementary Table 4.

We agree that FML017 does not appear to have incomplete penetrance and indeed this was a mistake on our part and our intention was to highlight FML018 instead. We are sorry for this error on our part.

In relationship to FML041, we have genotyped two of the maternal aunts (II-3 and II-10) and who are asymptomatic carriers for the *TERT* c.1445delA variant. This data is now shown in the family tree in Fig. 2k.

4. Page 1: other non-hematological features comment: these features should be added to Table 3 as described above to convince the reader that patient had the associated syndrome

The non-hematological features (such as deafness, short stature, leucoplakia, radioulnar synostosis) observed in the syndromic cases have been added to Table 3. We agree with the reviewer this additional information helps to convey the syndromic nature of some of these patients.

5. Page 2: “where genetic screening must venture outside the coding genome”: Confusing as stated. Could start that sentence at the “Germline RUNX1 deletions...” statement as seems more like emphasis is on CNV analysis rather than non-coding regions

This has now been amended.

6. Given that 32 of the 48 families with a primary genetic lesion identified were already known, a comment indicating that this series may not reflect yield of testing in fully uncharacterized familial mds/aml cases should be added.

We now state in the manuscript that our collections have been accrued over a significant period of time, by ‘word of mouth’ or ‘following publications’ and this will undoubtedly lead to some ascertainment bias, e.g. our historical interest in *CEBPA* and access to cases with a FPD-AML diagnosis in the absence of coding region mutations. This may well be less of a factor in the uncharacterized series of cases, where referral of cases would have been offered in the absence of any underlying genetic data.

Figure 1:

1. Would be more effective if presented as number of families with variants in the genes on the x axis sufficient for a clinical diagnosis of the respective syndrome. Those with VUS or heterozygous changes in genes in which homozygosity or compound heterozygosity is more convincingly diagnostic of a syndrome should be left out or highlighted in a different way so it is easier for reader to appreciate

Figure 1a shows the number of families (on the y-axis) with variants in the different disease genes (on the x-axis). We have added to the legend that variants in these genes are heterozygous apart from for *ERCC6L2*, *FANCA* and *SBDS* where they are biallelic and for *WAS* where it is hemizygous. Following our new classification of families in 2 groups (Table 1) on based on the panel used for clinical indication in the NHS England Genomic Medicine Service (<https://www.england.nhs.uk/publication/national-genomic-test-directories/>), we have also differentiated in the figure the three subgroups within Group 1 families: *High level of evidence for gene-disease association*, *Moderate level of evidence for gene-disease association* and *Genes emerging from basic research or mutated in other inherited hematological syndromes with high risk of MDS/AML*.

Figure 2:

1. pedigrees would benefit from age of onset of the hematologic malignancies

In the new Figure 2 we have now provided the age of onset of the hematological malignancies/diseases where this information is available.

2. clarify meaning of “NA” in the key or figure legend

Not available. This is now clarified in the legends.

Figure 3:

1. TP53 germline mutations are known to contribute to leukemia risk; this gene should really be in group1/2

We agree, and have moved them accordingly.

2. venn diagram also seems arbitrary in its divisions; how are DHX34, etc in familial mds/aml but DDX41 is considered a classical inherited BMF syndrome with constitutional thrombocytopenia? This diagram could be removed

The Venn diagram has been removed. Figure 3 is now a schematic representation of mutated genes in our cohort (Group 1 genes and selected new candidate genes) showing the overlapping between familial MDS/AML and hematological syndromes from a genetic point of view.

Table 1:

1. Group 1 versus Group 2 would benefit from more refined definitions. As currently presented, having genes like FANCA and SBDS for which approved testing exists and there is a clear predisposition to MDS/AML placed into Group 2 versus SAMD9 and SAMD9L in Group 1 (testing is harder to get for these new genes) is confusing. Consider separating into:

Group 1: Families with variants diagnostic of a known familial MDS/AML or inherited marrow failure syndrome (this would then include these Fanconi anemia and SDS cases); if exome sequencing was performed on all cases, then genes included in this category should absolutely be expanded (e.g. even TP53 is not on this list and is known cause of familial mds/aml)
Group 2: Families with variants in preliminary evidence genes. This category could then specify genes for which heterozygous variants were found in genes that homozygous or compound heterozygous mutations are more strongly associated with hematologic malignancy (e.g. RTEL1 or PARN)

The manuscript has been considerably lengthened, allowing us to describe in much greater depth the MDS/AML families, and the division of families. We now recognize that the division of families into 3 groups has caused confusion. We have now rationalized families into 2 discrete groups (Table 1) based on the panel used for clinical indication 'R347 Inherited predisposition to acute myeloid leukaemia (AML)' in the NHS England Genomic Medicine Service (<https://www.england.nhs.uk/publication/national-genomic-test-directories/>): Group 1 families (FML001-049; Supplementary Figure 1; Supplementary Table 1) where a variant has been detected either in an established locus or emerging from basic research but requiring more validation (Group 1 includes *TP53* and *SAMD9*); and Group 2 (FML050-086; Supplementary Figure 2; Supplementary Table 2), where the molecular lesion has not been defined.

Table 2:

1. Define family- history of hematological disease more precisely

In this study, familial AML/MDS refers to families where two or more family members (usually first degree) were diagnosed to have a hematological disorder (predominantly AML, MDS, aplastic anemia, thrombocytopenia) with at least one of the affected cases being categorized as MDS or AML. By definition therefore there is family history of hematological disease (AML, MDS, aplastic anemia or thrombocytopenia) in all these 86 families.

Table 3:

1. see comments above

We have added the VAF to Table 3 and expanded it considerably as a Supplementary Table 4, to include REVEL and CADD annotation, the discovery method, the tissue source and the disease status at time of sampling, as suggested above and in response to other reviewers' comments.

Reviewer #3 (Remarks to the Author):

The present study is led by a highly reputable group and focuses on the characterization of germline mutations in myeloid disease. The authors have in their hands a truly unique cohort. This study has the potential to significantly advance our understanding of the specific mutations and genes that contribute towards inherited risk for myeloid neoplasia. These findings can shape development of clinical diagnostic assays for the detection of germline predisposition variants, inform genetic counselling practices and uncover new biology. However, despite this significant resource, the manuscript suffers from lack of methodological detail as well as structure in the presentation of findings. A significant proportion of the relevant information is dispersed amongst the figures and

supplemental material. The paper in itself is hard to follow and put together. This precludes it from publication in its current form. The authors should consider investing some effort in presenting the samples, methods and notation of their findings before resubmission of this manuscript. A few recommendations for the manuscript are provided below:

The authors do not provide any detail on the results section on the number of individuals tested or ascertained. The authors do not discuss detailed methods for the annotation of each mutation. In the method's the authors mention under criterion (5) that "practical segregation amongst affected individuals was considered".

However, none of this information is provided in the text. How many families had more than 1 proband tested? Were any unaffected individuals tested? The authors should provide a very detailed overview of the number of families with 1, 2, 3 or more cases available for testing in the main text as well as supplemental material. The authors should provide information on the timing of sampling acquisition for molecular profiling and how they addressed biases from sampling time relative to manifestation of clinical phenotype.

We wholeheartedly agree with the reviewer that the manuscript would benefit from significant additional information, we have now included detailed information in a new table (Supplementary Table 3) and in the text and that the concerns raised with regards the 'lack of methodological detail as well as structure in the presentation of findings' has been remedied. We would want the reviewer to be aware that the editors of NC had agreed to review the manuscript, in a letter format, as submitted to a sister nature journal, on the understanding if the work was deemed of sufficient merit that we would amend and provide additional information at resubmission. Supplementary Table 3 includes information for all 86 families (FML001-086) on the number of (i) subjects, (ii) samples, (iii) sample type (iii) segregation data (iv) sequencing approaches and an overall summary of the collection.

In the methods the authors say that they performed targeted gene resequencing and prepared whole exome sequencing libraries (which is valid, as this reflects the right library preparation method for capture based assays). However, the authors then proceed in discussing analyses of whole exome sequencing data. The authors should clarify the nature of the assay and data generated for this project. The authors discuss 10 genes in the methods, but show more genes in both tables and main figures. The authors should be very specific on what assay was generated for how many individuals.

In hindsight, we recognize that the letter format of the initial submission, limited the opportunities to go into sufficient detail on many of the sample and technical approaches undertaken. This was an error that we now hope has been remedied in our revision. We have assigned our families into 2 discrete groups: Group 1 families (FML001-049; Supplementary Figure 1; Supplementary Data 1) where a variant has been detected either in an established locus or emerging from basic research but requiring more validation and Group 2 (FML050-086 supplemental Supplementary Figure 1; Supplementary Data 1), where the molecular lesion has not been defined. The sequencing and variant data for Group 1 genes were accumulated by a number of different sequencing approaches, that included testing using the accredited panel in the WMRGL, standard sanger sequencing and CNV profiling using SNP and CGH arrays. We very much hope that the reviewer recognizes the dynamic nature of the field at the present time and that cases will move from Group 2 to Group 1 as and when

new loci gain credence in the literature and multiple examples of families are detected. Indeed, a case in point is the example of *SAMD9L* variants which in our series were initially detected by WES.

The authors provide very little methodological detail on the post processing and filtering criteria of the data. How did they authors distinguish between rare CHIP/ARCH variants and rare germline events. For example, mutations in *TET2* are frequently seen CHIP/ARCH as clonal with low population based minor allele estimates. The authors discuss the detection of 4 putative germline variants in *TET2* as potentially new variants but do not discuss their methodology neither their validation of this observation. One validation of such events as germline would be to identify them in related and affected individuals. However, such data are referenced in the text, nor how these were specifically excluded from being CHIP variants.

We now detail the evidence we have available in each family to establish the fact that the variants identified are indeed germline. However, it is true that there are 25 families where this is not possible, for the reasons detailed above. More detailed methods and filtering criteria of the data are now included in the manuscript. In the case of our four *TET2* families (FML054, FML051, FML075 and FML081), we have demonstrated segregation in FML054 (p. Ala241Val) where DNA from patient I.1 was available (validation now in Supplementary figure 5). There is enough evidence to consider FML051 as a *TP53* family and is now included in our Group 1 families, suggesting that the identified *TET2* variant is not the germline disease-causing event. Only the index case sample was available in the other two families with *TET2* mutations (FML075, p.Cys1464*, VAF=0.46; and FML081, p.Pro1962Leu, VAF=0.41) and we accept the point that may be CHIP variants. We have nuanced our description of these *TET2* variants accordingly in the manuscript. The challenges faced in confidently predicting a variant as pathogenic is now discussed in detail in the discussion, as it reflects one of the major challenges in the identification of the loci that can be linked to Group 2 families.

Please provide a detail summary in the main text of total number of families, affected individuals and non-affected individuals (criteria for non-affected) for who you ascertained samples for this study.

This information is now detailed in our new sample table (Supplementary Table 3) and documented in the first part of the result section.

When providing summary information on the germline data, please include information on whether the precise mutation identified has been previously observed and validated as germline as opposed to representing a previously unseen. Also include in supplemental material the categorization of previously known versus previously unseen mutation as mapped by 1 affected individual, 2, 3 or more.

We agree with the reviewer that discriminating germline from acquired and indeed pathogenic from passenger variants is of critical importance. Our collections have been accrued over a significant period of time, by 'word of mouth' or 'following publications' where both national and international collaborators (13 countries included in the study) have provided material for analysis. The availability of fibroblasts or sorted T cells, which would provide a gold standard approach to discriminate germline from acquired is not practical in our series, given the historical nature of our collections. That said, we had an opportunity to test many variants across >1 affected individuals, the use of remission or

buccal DNA or lastly through *in silico* analysis and document the frequency of these variants in the ExAc and gnomAD databases. While not confirmatory by any means the median VAF of all 130 of our selected variants was 0.48.

The authors discuss functional validation but do not show any functional such data or discuss it in any detail in the text.

We have not provided functional data for the novel variants in the known Group 1 disease genes as there is an acceptance that these are all *bona fide* loci, have been functionally validated and are recurrently mutated in familial AML/MDS. Instead we have focused attention on one of the most frequently mutated candidate genes in our series, the RNA helicase *DHX34* and working in conjunction with the leading group worldwide in this field, have shown that all 4 gene variants detected in 4 families, compromise NMD activity. This sections describing both the clinical features of the *DHX34* families and the functional assessment of these variants have been expanded both in the results and discussion sections of the manuscript.

Did the authors have access to diagnostic samples? It would be interesting to test whether at diagnoses which of these variants often lose the wild type allele. Such an observation could serve as a further validation or evidence of the putative oncogenic role of such mutations.

This is a valuable suggestion and we agree entirely that the co-occurrence of a germline variant and an acquired variant (e.g. as in examples of *CEBPA* and *DDX41*) would offer an important validation. However such biallelic inactivation, may well be the exception rather than the rule. In our analysis on diagnostic samples, the majority of the known dominant disease genes, did not demonstrate loss of the wild type allele and the VAF was consistently close to 0.5 which is now shown for all variants in Table 3 as well as the Supplementary Table 6). We know from some recent data from our group that other mechanisms, aside say mutation or LOH may well be important; in our recent studies on a *GATA2* pedigree (Al Seraihi et al., Leukemia 2018) we demonstrated allele specific expression of the mutant allele by silencing of WT expression.

Appended information

In order to respond as fully as possible to the reviewers' comments, we have changed the format of the manuscript from letter to article enabling detailed expansion of technical and clinical aspects. We have also edited and rearranged the tables and figures and we thought it might be helpful to summarize these edits as follows:

1. Figure 1 has been modified to accommodate the new grouping of the families.
2. Figure 2 has been modified to include a TP53 family (which was FML051 is now FML049). The age of each individual at presentation (where available) has been added to this figure in grey italic.
3. Figure 3 has been modified to accommodate reviewer's comments: Fig. 3a and Venn diagram from Fig. 3b have been removed.
4. Figure 4 has been modified to accommodate reviewers' comments by repeating the corresponding experiment and adding a new panel d on Figure 4.
5. Table 1 has been modified to accommodate the new grouping of the families.

6. Table 2 remains the same.
7. Table 3 has been extended to include the variant allele frequency of the variants for which that data is available and non-hematological features associated to the families. Footnote showing telomere length measurement has been also added to this table.
8. Supplementary figures have been merged in a single file following Nature Communication guidelines.
9. Supplementary Figure 1 has been modified to accommodate the new grouping of the families.
10. Supplementary Figure 2 has been incorporated into Supplementary Figure 1 (Group 1 families).
11. Supplementary Figure 3 has become Supplementary Figure 2 and been modified to accommodate the moving a TP53 family to Supplementary Figure 1.
12. Supplementary Figure 4 has become Supplementary Figure 3.
13. Supplementary Figure 5 has become Supplementary Figure 4 and been modified to accommodate the reduced number of candidate genes.
14. Supplementary Figure 6 has been modified to now show that all 4 *DHX34* variants display a decreased recruitment of the NMD factor UPF2 and UPF3b to UPF1. We are also providing uncropped Western blots for this Figure and all other Western Blots as part of Source data.
15. Supplementary Figure 7 has become Supplementary Figure 5 and we have added the sequencing traces for all 47 variants selected and validated by Sanger sequencing. It had also one TP53 variant removed, which was found not to segregate.
16. Supplementary Tables 1 and 2 have been fused, and called Supplementary Table 1 (Group 1 families).
17. Supplementary Table 3 becomes Supplementary Table 2 (Group 2 families).
18. A new Supplementary Table 3 has been created, detailing all the samples available for the study.
19. A new Supplementary Table 4 has been created, to provide further detail to Table 3 of the main text.
20. Supplementary Table 4 has become Supplementary Table 6. It has been modified as follows:
 - a. Columns have been added to detail the CADD and REVEL scores of all the variants.
 - b. Columns have been added to show read depth and variant allele frequencies
 - c. Eight genes have been removed, resulting from further segregation studies and the movement of a TP53 family to Group 1.
21. Supplementary Table 5 has been modified removing one FML051 synonymous variant.
22. Supplementary Table 6 becomes Supplementary Table 7.
23. Supplementary Table 7 becomes Supplementary Table 8.
24. Supplementary Table 8 becomes Supplementary Table 9 and we have added the primers corresponding to all 47 variants selected and validated by Sanger sequencing.

We have also made the following changes as a result of moving family FML051 into the Group 1 families as well as further segregation analysis:

1. FML056 (TP53:Arg273His) has been removed from Supplementary Table 7, as this variant was shown not to segregate.
2. FML051 (TP53:Arg282Trp) has been removed from Supplementary Tables 3, 5 & 7 and placed in Supplementary Table 1 (Group 1 families), where it becomes FML049. To compensate for this change, FML049 and FML050 become FML050 and FML051 respectively.

3. Moving FML051 to Supplementary Table 1 (Group 1 families) results in the loss of five of the candidate genes in Supplementary Table 6 (*CNTN2*, *KRTAP10-6*, *MEP1A*, *MMP10* and *PDCD4*).
4. Further studies have shown that an additional 10 variants (in *ABL2*, *ADA*, *APOB*, *COL5A1*, *DNAH9*, *IL17RA*, *NPHS1*, *TCHP*, *TET2* and *VWWDE*) are segregating, confirming their germline status. This detail has been added to Supplementary Table 6. However, three of the candidate genes were found not to segregate and were removed from Supplementary Table 6 (*DNAH11*, *MLTK* and *TLDC1*).

Reviewers' comments:

Reviewer #1 (Remarks to the Author):

I compliment the authors on this revision. Although numerous, they sufficiently addressed my comments and concerns. This is very strong manuscript.

Reviewer #3 (Remarks to the Author): Also asked to check your rebuttal to Reviewer #2 who was unable to re-review

The revised manuscript by Rio-Machin and colleagues has significantly improved, particularly with regards to the cohort and methodological description. However, a few outstanding issues remain.

1. In the abstract the current wording of "52 new candidate genes" without explanation of how many retained candidacies within this study appears misleading. It will be worthwhile to discuss 52 candidate genes of which x we could define a putative role in germline pathogenesis. This includes etc. The abstract should also include a final categorization of families into known, putative and unknown/uncharacterized cases. Do the authors support a causal link for all 52 genes?
2. In the description of the three FPD-AML cases, the authors discuss large copy number variants detected but do not provide information about the locus, the size of the variants and the segregation in the text. Supplementary figure 3 shows copy number analyses of RUNX1 exons using MLPA. In the methods the authors discuss that derivation of copy number data was conducted using CGH array and MLPA. Given their statement in the text "highlighting the importance of performing copy number assessment" the authors provide little methodological detail or guidelines of how copy number analyses in familial cases can support the diagnostic work up of cases with suspected familial disease. Importantly both targeted gene sequencing and exome sequencing offer the opportunity for systematic evaluation of copy number alterations with well-established methods. Did the authors perform such analyses using the targeted gene sequencing or exome data to a. validate the ascertainment of the RUNX1 events, and b. complement their study with a global copy number analyses across all candidate loci? It would be very helpful if methodological detail and further clarification was provided on this point.
3. How did the authors choose the EXAC (0.0001) cutoff? Did the authors evaluate variants in candidate loci that may be present above this threshold? Additionally, the authors impose a limit in variant selection on the basis of a gene being mutated in 2 families or more. Given the small size of the discovery cohort (n=37), a gene mutated in at least 2 variants would account for at least 5% of the residual risk. This may be unattainable given the rarity of many of the low penetrant germline predisposition genes. Did the authors also consider evaluating any mutations in genes implicated in somatic mutation in hematological disorders (neoplasms/bone marrow failure etc.) and or other familial predisposition genes irrespective of frequency in EXAC and or the cohort? For example, beyond the restricted criteria did the authors cross reference the starting candidate gene list with the genes in Bluteau et al, Churpek et al etc.?
4. Where the authors state that multiple variants were detected in the majority of the families, can they provide median and range?
5. The authors state that 12 of the 52 genes were previously associated with examples of sporadic AML but do not provide any information on the remaining 40.
6. In the section "Rationalizing candidate genes into discrete groups". Do the authors elude to the 52 genes? It would be helpful for the reader if the authors were more explicit in the text i.e. We next sought to rationalize these 52 candidate genes by dividing them into ... This section lacks a summary overview or conclusion. How did the authors handle the TE2 cases? Given that many families had multiple variants (as mentioned in the previous section) was any overlap observed? Where there any differences in transmission, age of onset etc. amongst the different gene family types?
7. For DHX34 families. It is hard to believe that the families did not have further secondary lesions

in the presence of limited profiling. The authors should either expand in the methodology and data that support this conclusion (provide cytogenetic/ karyotype information for the cases that developed AML plus methodological and mutation data in the 33 genes listed. In the absence of detailed and convincing data to support this statement I would suggest to be conservative with this statement as that would imply very strong effects for DHX34 in disease pathogenesis.

In conclusion, this work represents a commendable amount of work in a unique cohort of familial ads/aml cases. Beyond the cataloguing of mutations, the manuscript is limited in presenting a global overview, significance and novelty of the data as well as lessons learned from this duty with regards to diagnostic testing, disease surveillance or sample acquisitions. In the absence of the latter, a simple cataloguing of the mutations is limited in scope. An effort towards a global description of the dataset beyond the molecular groups would significantly strengthen the manuscript.

Reviewers' comments:

Reviewer #1 (Remarks to the Author):

I compliment the authors on this revision. Although numerous, they sufficiently addressed my comments and concerns. This is very strong manuscript.

Reviewer #3 (Remarks to the Author): Also asked to check your rebuttal to Reviewer #2 who was unable to re-review

The revised manuscript by Rio-Machin and colleagues has significantly improved, particularly with regards to the cohort and methodological description. However, a few outstanding issues remain.

1. In the abstract the current wording of “52 new candidate genes” without explanation of how many retained candidacies within this study appears misleading. It will be worthwhile to discuss 52 candidate genes of which x we could define a putative role in germline pathogenesis. This includes etc. The abstract should also include a final categorization of families into known, putative and unknown/uncharacterized cases. Do the authors support a causal link for all 52 genes?

We thank the reviewer for their comments and we have now included a sentence in the abstract (lines 81-83) to improve the overall clarity of presentation.

2. In the description of the three FPD-AML cases, the authors discuss large copy number variants detected but do not provide information about the locus, the size of the variants and the segregation in the text. Supplementary figure 3 shows copy number analyses of RUNX1 exons using MLPA. In the methods the authors discuss that derivation of copy number data was conducted using CGH array and MLPA. Given their statement in the text “highlighting the importance of performing copy number assessment” the authors provide little methodological detail or guidelines of how copy number analyses in familial cases can support the diagnostic work up of cases with suspected familial disease. Importantly both targeted gene sequencing and exome sequencing offer the opportunity for systematic evaluation of copy number alterations with well-established methods. Did the authors perform such analyses using the targeted gene sequencing or exome data to a. validate the ascertainment of the RUNX1 events, and b. complement their study with a global copy number analyses across all candidate loci? It would be very helpful if methodological detail and further clarification was provided on this point.

The reviewer raises an important point and we would want to reassure them that, we have worked hard in characterizing these FPD-AML families with *RUNX1* deletions. Furthermore, in one case (FML031) we were also able to narrow down the breakpoint region and create a variant specific PCR that could be used for diagnosis purposes and demonstrated a *RUNX1* deletion of 666,561bp whose co-ordinates varied only 175bp from the breakpoint predicted by CGH-array (see figure below). In response to the reviewer, we have amended the corresponding sections in the Results (lines 177-191) and Methods (lines 451-461), clarifying the methodological procedures applied to detect *RUNX1* deletions in the three families with FPD-AML phenotype without germline *RUNX1* variants. Moreover, we have created a new supplementary figure (Supplementary figure 3a) showing Array CGH and SNP-

array results for the corresponding families. These three families were not submitted to WES, as these germline *RUNX1* deletions had already been detected using one discovery and one validation method (CGH-array/SNP-array and MLPA). No other families with clinical features of FPD-AML and the absence of germline nucleotide variant were identified in our cohort and we chose not to follow a bioinformatics strategy to search for CNVs from WES/gene panel data (e.g. ExomeDepth) as in our hands they are not sufficiently robust and tend to throw up many false outputs.

Figure: *RUNX1* deletion detected in family FML031. Array CGH showing germline *RUNX1* deletion (A) with Sanger sequencing identifying the precise breakpoint of the deletion (B).

3. How did the authors choose the EXAC (0.0001) cutoff? Did the authors evaluate variants in candidate loci that may be present above this threshold? Additionally, the authors impose a limit in variant selection on the basis of a gene being mutated in 2 families or more. Given the small size of the discovery cohort (n=37), a gene mutated in at least 2 variants would account for at least 5% of the residual risk. This may be unattainable given the rarity of many of the low penetrant germline predisposition genes. Did the authors also consider evaluating any mutations in genes implicated in somatic mutation in hematological disorders (neoplasms/bone marrow failure etc.) and or other familial predisposition genes irrespective of frequency in EXAC and or the cohort? For example, beyond the restricted criteria did the authors cross reference the starting candidate gene list with the genes in Bluteau et al, Churpek et al etc.?

We are very grateful to the reviewer as he/she has perfectly articulated the challenges faced in rationalizing the large volume of variants arising in WES analysis from what in our understanding represents the largest set of discovery cases reported to date. We are very conscious, as indeed is the reviewer based on their reviews that the precise number and nature of individual variants will differ depending on the criteria chosen. We have therefore spent significant time in outlining our criteria by analyzing all the germline variants in *bona fide* disease genes described in the literature so far in familial MDS/AML (e.g. most of these variants are not present in population controls and are predicted to be pathogenic with the different tools used for functional annotation).

As requested by the reviewer we have now gone back to our starting candidate list without filtering (9,468 variants in 6,637 different genes) and crossed reference with the candidate genes presented in the studies of Bluteau et al and Churpek et al. This analysis allows us to highlight two further genes, *MDM4* mutated in FML084 and *PRF1* mutated in FML085 (just 1 family each) that also meet our criteria (frequency of <0.0001 in population controls (ExAC database) and predicted to be pathogenic in at least two out of four tools used for functional annotation) and these are now incorporated into the Supplementary Table 8. In response to the reviewer's comments, we have also created a

Supplementary Table 8b for revision purposes with the list of variants in genes, common to Bluteau et al and Churpek et al manuscripts that do not meet our criteria. There are 28 variants in 15 genes in 14 families where based on our pipelines, 18 of these 28 variants correspond to supposed SNPs. We have also cross referenced the starting candidate list of 9,468 variants with the list of 492 “essential” genes in AML described by Tzelepis et al (2016, Cell Reports) after their CRISPR screening, and we identified 149 genes in common; what these data demonstrate, is the need to incorporate an extra level of filtering to enrich the final list of candidates to the most likely variants/genes. Certainly, this approach will lead to the exclusion of *bona fide* loci and hence the need to be as transparent as possible in the assumptions and criteria set in our analyses.

In this study, and we hope that the reviewer recognizes, that we further rationalized our list of 52 candidate genes, to focus attention on seven genes with a putative role in germline pathogenesis. And critically, in order to both inform and progress this area of study, we have also provided the raw data in order to allow researchers working in the field to cross reference with the list of variants identified in their own cohorts of families. We consider that is the best way to rationalize variants and define new *bona fide* disease genes to be included in diagnostic sequencing panels. Reflecting on this ambition, we would ask the reviewer to take another look at the discussion section of the manuscript, where we consciously refrained from making overly profound statements but instead we believe, our new data and insight offer a perspective on the challenges inherent in the field and a framework for other groups to build knowledge in this important area of myeloid research.

4. Where the authors state that multiple variants were detected in the majority of the families, can they provide median and range?

According to the referee’s suggestion we have now added the average and range (line 327).

5. The authors state that 12 of the 52 genes were previously associated with examples of sporadic AML but do not provide any information on the remaining 40.

We hope that the reviewer recognizes that the weight of evidence supporting the selection of 12 genes, will depend on the nature of the literature we have selected for comparison. Again, we have been mindful to describe the path taken in making these comparisons. To identify candidate genes also mutated in sporadic AML, in this revised version, we have also included data from 672 AML patient presented by Tyner et al (2018, Nature) and 20 of the 52 genes have been reported to be mutated in at least one patient with sporadic AML from these series. Supplementary figure 4 and lines 227-229 from the manuscript have been modified accordingly. Regarding the remaining 40 genes, their function and import in other diseases varies. Therefore, we have included an additional four columns in Supplementary table 6 indicating whether there is an **associated inherited disorder** linked to the specific gene, **association with leukemia and other cancers**, the **function** of the protein and whether these genes have been reported to be **mutated in sporadic AML** (using TCGA and Tyner et al. data).

6. In the section “Rationalizing candidate genes into discrete groups”. Do the authors elude to the 52 genes? It would be helpful for the reader if the authors were more explicit in the text i.e. We next sought to rationalize these 52 candidate genes by dividing them into ... This section lacks a summary overview or conclusion. How did the authors handle the TE2 cases? Given that many families had multiple variants (as mentioned in the previous section) was any overlap observed?

Where there any differences in transmission, age of onset etc. amongst the different gene family types?

We have now amended the starting sentence of this section according to the referee's comment (line 232) and we have added a paragraph clarifying the nature of the *TET2* mutated cases (lines 256-259). Although we found co-occurrences of germline variants in some families, we have not observed any particular correlation in clinical manifestations and disease progression.

7. For DHX34 families. It is hard to believe that the families did not have further secondary lesions in the presence of limited profiling. The authors should either expand in the methodology and data that support this conclusion (provide cytogenetic/ karyotype information for the cases that developed AML plus methodological and mutation data in the 33 genes listed. In the absence of detailed and convincing data to support this statement I would suggest to be conservative with this statement as that would imply very strong effects for DHX34 in disease pathogenesis.

Accordingly with the reviewer's suggestion, we have now included karyotype information (available for 2 families and mentioned earlier in the section) in lines 283-284. In order to be as transparent as possible, we now include the list of 33 genes analyzed (lines 413-418) and additional methodological information has been added to the corresponding Methods section (lines 420-423).

Reviewers' comments:

Reviewer #3 (Remarks to the Author):

The authors present a significantly improved version of the manuscript. A few outstanding issues remain:

1. This manuscript would be strengthened by a systematic copy number analyses using established tools such as the Germline CNV caller from GATK but I understand if it cannot be accommodated for this manuscript.
2. A recurrence of 3 families remains stringent, the authors should consider relaxing that criteria and providing a decision tree.
3. Consolidation of variant information into a single supplementary table would significantly support dissemination of findings, as well as an evaluation of the data, methodology and main findings. It would therefore be very important to create a single table that contains the following sections:
 - a. Variant information: Chromosome, position, reference allele, alternative allele VAF (and reference genome build).
 - b. Segregation information within family)
 - c. Recurrence within cohort overall
 - d. Population based metrics (as currently shown)
 - e. Predicted variant annotation per standard tools (as currently shown)
 - f. Membership in one of the 3 criteria groups / lists.
 - g. Final assignment consideration by the authors and any comments
4. In the absence of further supporting evidence the two TET2 variants in families 259 and 261 should be annotated as most likely CHIP/ ARCH.
5. ST9: Should include chromosome co-ordinates and built

Reviewer #3 (Remarks to the Author):

The authors present a significantly improved version of the manuscript. A few outstanding issues remain:

We thank the reviewer for their kind words and are gratified to hear that the manuscript has significantly improved, following the recent revision.

1. This manuscript would be strengthened by a systematic copy number analyses using established tools such as the Germline CNV caller from GATK but I understand if it cannot be accommodated for this manuscript.

We agree with the reviewer the CN profiles would be a valuable inclusion. However, as mentioned previously, in our hands the use of WES data/CNV caller is suboptimal. That being said, the strength of the manuscript is that we have made available all the primary data so that CN analysis can be interrogated further by the wider research community.

2. A recurrence of 3 families remains stringent, the authors should consider relaxing that criteria and providing a decision tree.

We have now created a new Supplementary Figure 4 (decision tree) and modified the filtering applied to our WES data. The filtering process is clarified in the manuscript (lines 213-228 and lines 241-243) and referenced in Figure 3 and Supplementary Figure 5 accordingly. As suggested by the reviewer, we have now relaxed the criteria of *number of families* for genes related to rare hematological syndromes and common to other series of cases. Therefore, a new gene *PRF1*, mutated in one of our families and in one described by Bluteau *et al.* and associated with Familial hemophagocytic lymphohistiocytosis-2 development, is now included as a likely causal gene in our final candidate list (Supplementary Table 8 and lines 250-251 and 343). We have also added for completeness another validated gene, *IL17RA*, linked to a rare hematological syndrome (Immunodeficiency-51) (lines 249-250). *PRF1* and *IL17RA* germline mutations were validated by Sanger sequencing (Supplementary Fig. 6). Overall, these amendments by the reviewer, has led to an increase in the number of candidate genes to 65, now shown in supplementary Figure 8.

For the purpose of this study, we have established our filtering criteria based on the *bona fide* disease variants that have been identified previously in familial MDS/AML. It is worth restating that the final list of candidate genes will vary as the criteria are modified. There is however an opportunity for the

scientific community to apply their own criteria to the raw WES data, as this is now provided in the manuscript (EGAD00001004539).

3. Consolidation of variant information into a single supplementary table would significantly support dissemination of findings, as well as an evaluation of the data, methodology and main findings. It would therefore be very important to create a single table that contains the following sections:

In agreement with the reviewer we have now created a new Supplementary table 8 including all the requested sections:

- a. Variant information: Chromosome, position, reference allele, alternative allele VAF (and reference genome build).** Done
- b. Segregation information within family:** Done. Segregation has been validated in all the variants when DNA from 2 or more members of the family was available. Members analysed for every variant are showed in column A (*Family (patient)*).
- c. Recurrence within cohort overall:** Done. The table is ordered by gene, showing the list of families with variants in each gene.
- d. Population based metrics (as currently shown).** Done
- e. Predicted variant annotation per standard tools (as currently shown).** Done
- f. Membership in one of the 3 criteria groups / lists.** Done (column AC, *Criteria Group*)
- g. Final assignment consideration by the authors and any comments.** Done (Column AH, *Comments*)

4. In the absence of further supporting evidence the two TET2 variants in families 259 and 261 should be annotated as most likely CHIP/ ARCH.

TET2 variants in families 259 and 261 are now annotated as *most likely CHIP/ ARCH* in Supplementary Table 8.

5. ST9: Should include chromosome co-ordinates and built

Supplementary Table 9 now includes chromosome co-ordinates and built for each pair of primers.